

# Evaluation of remote sensing and reanalysis based precipitation products for agro-hydrological studies in semi-arid tropics of Tamil Nadu

Aatralarasi Saravanan[1,2], Daniel Karthe[1,2,3], Selvaprakash Ramalingam[4] and Niels Schütze[2]

[1]United Nations University – Institute for Integrated Management of Material Fluxes and of Resources (UNU-FLORES), Dresden, 01067, Germany
[2]Faculty of Environmental Sciences, Technische Universität Dresden, Dresden, 01069, Germany
[3]Faculty of Engineering, German-Mongolian Institute for Resources and Technology (GMIT), Nalaikh, Mongolia
[4]Division of Agricultural Physics, Indian Agricultural Research Institute, New Delhi, 110012, India

*Correspondence to*: Niels Schütze (niels.schuetze@tu-dresden.de)

**Abstract.** This study provides a comprehensive evaluation of eight high spatial resolution gridded precipitation products in semi-arid regions of Tamil Nadu, India, focusing specifically on Coimbatore, Madurai, Tiruchirappalli, and Tuticorin, where both irrigated and rainfed agriculture is prevalent. The study regions lack sufficiently long-term and spatially representative observed precipitation data, essential for agro-hydrological studies and better understanding and managing the nexus between food production and water and soil management. Hence, the present study evaluates the accuracy of five remote sensing-based precipitation products, viz. Tropical Rainfall Measuring Mission (TRMM), Precipitation Estimation from Remotely Sensed Information using Artificial Neural Networks – Climate Data Records (PERSIANN CDR), CPC MORPHing technique (CMORPH), Global Precipitation Measurement (GPM) and Multi-Source Weighted-Ensemble Precipitation (MSWEP) and three reanalysis-based precipitation products viz. National Center for Environmental Prediction  Reanalysis 2 (NCEP2), and European Centre for Medium-Range Weather Forecast (ECMWF) Reanalysis version 5 Land (ERA5-Land), Modern-Era Retrospective analysis for Research and Application version 2 (MERRA 2) against the station data. Linearly interpolated precipitation products were statistically evaluated at two spatial (grid and district-wise) and three temporal (daily, monthly, and yearly) resolutions for 2003-2014. Based on overall statistical metrics, ERA5-Land was the best-performing precipitation product in Coimbatore, Madurai, and Tiruchirappalli, with MSWEP closely behind. In Tuticorin, however, MSWEP outperformed the others. On the other hand, MERRA2 and NCEP2 performed the worst in all the study regions, as indicated by their higher Root Mean Square Error (RMSE) and lower correlation values. Except in Coimbatore, most precipitation products underestimated the monthly monsoon precipitation, which highlights the need for a better algorithm for capturing the convective precipitation events. Also, the Percent Mean Absolute Error (%MAE) was higher in non-monsoon months, indicating that these product-based agro-hydrological modeling, like irrigation scheduling for water-scarce periods, may be less reliable. The ability of precipitation products to capture the extreme precipitation intensity differed from the overall statistical metrics, where MSWEP performed the best in Coimbatore and Madurai, PERSIANN CDR in Tiruchirappalli, and ERA5-Land in Tuticorin. This study offers crucial guidance for managing water resources in agricultural areas, especially in





precipitation data-scarce regions, by helping to select suitable precipitation products and bias correction methods for agro-hydrological research.

## 1 Introduction

Tamil Nadu, an agriculturally important state in India, is endowed with only 3 percent of the country's water resources. Of this limited supply, more than 95 percent of surface water and 80 percent of groundwater are already allocated for various uses, including human and animal consumption, irrigation and industrial activities (Anonymous, 2024). The state, which is in the rain shadow region of the Western Ghats, receives an average annual precipitation of about 925 mm. Agriculture relies heavily on precipitation and provides livelihood to nearly 40 percent of the people. The gross sown area is 5.94 million ha, while the net sown area is 4.73 million ha. The total irrigated area in the state is 3 million hectares; hence, around 50% of the gross sown area depends on precipitation as its primary water resource. In recent years, the state's agriculture sector has grown around 3% annually while its overall economy has risen by 6-9% (Government of Tamil Nadu, 2022). The decreased performance of the agricultural sector is collectively attributed to multiple factors like water shortages during critical stages of crop irrigation, low level of crop diversification, increasing climate change threats, the rigid mindset of the farmers in refusing to accept the improved technologies, weak market development, and high rates of post-harvest losses. Compared to other factors, climate change has differential impacts on the region's monsoons (Lal, 2016). Climate change has led to a noticeable rise in the intensity of rainfall in the months of August, October, and November, along with greater variation. For the agricultural sector, the most important implication of climate warming is an increase in irrigation water requirement, which is estimated to range between 0.6 to 3.7% for temperature increases between 0.5 to 3.0°C (Surendran et al., 2021). In addition to irrigated crops, climate-resilient rainfed crops, which include cotton, pearl millet, and sorghum, are expected to produce less by 2030 in all of Tamil Nadu's agroclimatic zones. Significant reductions are expected in the Northeast zone (6.07%), the Cauvery delta zone (3.55%), and the South zone (3.54%) (Arumugam et al., 2015). Furthermore, there has been an increase in the frequency of extreme hydrological events in the semi-arid region of the state, including the floods in Chennai in 2015 and 2022 and the hydrological droughts in 2018 caused by the failure of the northeast monsoon. Besides that, the floods that occurred in the Tuticorin district during December 2023 show that the state's extreme hydrological events are spreading to newer regions. Given the IPCC's projection of an increase in heavy precipitation events linked to tropical cyclones and worsening drought in certain regions, hydrological modeling plays a critical role in early warning systems. This modeling lays the foundation for the development of suitable environmental policies, which are essential given the state's current inconsistencies in warming, climate change, and rainfall impacts.

Establishing dense meteorological observatories and disseminating the state of weather variables at frequent intervals is essential for adapting to the regional and local consequences of climate change (Wilby and Yu, 2013). Developing countries like India are more vulnerable in this regard due to inadequate ground weather stations and, thus, a lower adaptation capacity. The distribution of rain gauges in one of the largest countries, India, is inadequate to accurately predict and assess various rain-





induced events/processes, like droughts, flash floods and their implications, such as dam failures (Singh et al., 2018). Despite the accurate measurement of rainfall by rain gauges, their heterogeneous distribution over land and relatively sparse measurements over large oceans limit their use. Also, gauge data are available as point observations with potentially limited spatial representativeness; inconsistencies exist between other data products, which is largely due to a limited number of ground stations, merging and interpolation methods, limited time resolution and limited documentation quality (Huffman et

al., 2009; Nikulin et al., 2012; Sylla et al., 2013) . The gaps and inhomogeneities in precipitation time series collected by the rain gauges at the time of severe weather conditions also make it difficult to force hydrological models that require a continuous time series of precipitation data.

The alternate solution is multi-source merged precipitation in general, including ground-based weather radars, meteorological satellites, and reanalysis products. These provide a comprehensive map of rainfall at homogeneous spatial and temporal

resolutions (Kucera et al., 2013; Tapiador et al., 2004). Several satellite-based rainfall estimates have been developed over the last decade (Sapiano and Arkin, 2009; Zambrano-Bigiarini et al., 2017). For India, a list of precipitation products is available that can be used for climate change impact studies, such as the Global Precipitation Climatology Project (GPCP) from the National Centers for Environmental Information of the National Oceanic and Atmospheric Administration (NOAA) with a spatial resolution of 2.5° (Adler et al., 2003), CPC Merged Analysis of Precipitation (CMAP) from the National Center for

Atmospheric Research (NCAR) with a spatial resolution of 2.5° (Xie et al., 2007), Tropical Rainfall Measuring Mission (TRMM) developed from the joint space mission between National Aeronautics and Space Administration (NASA) and Japan's National Space Development Agency with a spatial resolution of 0.25°(Climate Data Guide, 2024). In addition, Global Satellite Mapping of Precipitation (GSMaP) (Ushio et al., 2009), Precipitation Estimation from Remotely Sensed Information using Artificial Neural Networks – Climate Data Records (PERSIANN CDR) (Ashouri et al., 2015), Precipitation Estimation

from Remotely Sensed Information using Artificial Neural Networks – Cloud Classification System (PERSIANN CCS) (Sorooshian et al., 2000), CPC MORPHing technique(CMORPH) (Joyce et al., 2004), Global Precipitation Measurement (GPM) (Hou et al., 2014) and Multi-Source Weighted-Ensemble Precipitation (MSWEP) (Beck et al., 2017) are also available with at varying spatial and temporal resolutions for longer periods.

As another source of climate information, climate model-derived reanalysis data are suitable for assessing climate

variability and change. Global reanalysis-based climate products include National Center for Environmental Prediction Reanalysis 2 (NCEP2) (Kanamitsu et al., 2002), European Centre for Medium Range Weather Forecast (ECMWF) Reanalysis version 5 Land (ERA5-Land), 20th Century Reanalysis version 2 (20CRv2) (Compo et al., 2011), Climate Forecast Reanalysis System (CFSR) (Saha et al., 2010), Japanese 55-year Reanalysis (JRA-55) (Ebita et al., 2011), and Modern-Era Retrospective analysis for Research and Application version 2 (MERRA 2) (Gelaro et al., 2017).

Precipitation products from non-station sources must be assessed against ground station data before being imported into the Agro-hydrological models. The differential statistical assessment of the product at multiple spatial and temporal scales will highlight the accuracy of the precipitation product, quantify bias and lead to choosing the appropriate bias correction methods before forcing them into a hydrological model. Further, previous studies have shown the differential performance of





precipitation products in capturing extreme precipitation events, as well as their spatial variability. Hence, assessing these unique criteria is important to reduce the input uncertainty of hydrological modeling. Also, the precipitation data must be available for extended periods (at least 10 years) to study the impact of climate change on the hydrological cycle. The best-performing precipitation products when fed into Agro-hydrological models like the AquaCrop, Soil and Water Assessment Tool (SWAT, Neitsch et al., 2002), MODFLOW (McDonald and Harbaugh, 1988), Water evaluation and Planning (WEAP), Revised Universal Soil loss equation (RUSLE, Renard, 1997) can provide accurate projections with lower uncertainty. The thorough assessment of various precipitation products requires both dense station data availability and continuous time series of precipitation data for longer periods of time. Such assessment studies are not available for Tamil Nadu, particularly for the semi-arid regions on a daily timescale, which is crucial for comprehending hydrological variability in the context of climate change.

However, a limited number of studies at daily intervals are available based on the river basin spatial scale (Kolluru et al., 2020; Yaswanth et al., 2023). Further, (Singh et al., 2018) analyzed only heavy rainfall occurrences in peninsular India during the 2015 winter monsoon, taking grid-level spatial scale into consideration. Other investigations were carried out either at weekly or monthly scales (Dubey et al., 2021; Singh et al., 2018).

Thus, the purpose of this study is to compare and assess, using the most widely applied and recognized statistical and graphical evaluation methods, the available precipitation products for Coimbatore, Madurai, Tiruchirappalli, and Tuticorin at the highest possible spatial and temporal (daily, monthly, and yearly) resolution against station data. The results of our study will help to overcome the precipitation data scarcity in the study area with regard to the spatial and temporal resolution gaps of daily, monthly, and annual precipitation data products that can be used for hydrological and environmental change and impact studies at a regional scale.

## 2 Study area and data

### 2.1 Study region

The study's main objective is to assess precipitation data sources for semi-arid regions of Tamil Nadu (SAT), specifically Coimbatore, Madurai, Tiruchirappalli and Tuticorin, on a daily, monthly, and yearly basis. The study regions are representative of the following agroclimatic zones and are known to contribute significantly to the state's agricultural output: the Cauvery Delta Zone (Tiruchirappalli), the Western Zone (Coimbatore), and the Southern Zone (Madurai, Tuticorin) (Chandrasekar et al., 2009; Rajkumar et al., 2020). The state of Tamil Nadu receives precipitation during North-East monsoon (44%), South-West monsoon (41.27%), Hot weather season (11.41%) and Winter season (2.7%) (Dhar et al., 1981; Lakshmi et al., 2021). The study region's precipitation pattern is described in Table 1. The land surface heterogeneity, the intricate topography, and their interactions with global climatic forcing systems are some of the local elements contributing to the variability in the precipitation pattern in these locations. Similarly, weather and climate variability, both geographical and temporal, as well as tendencies toward rising temperatures and falling precipitation, are experienced throughout the study's





investigated locations. Additionally, research indicates that the rainfall window for the Southwest monsoon is getting shorter, while the northeast monsoon crops are vulnerable to flooding during their early stages (Varadan et al., 2017). Climate change has already started to change rainfall pattern in a way that is unfavorable for most agricultural crops, so measures for sustainable adaptation and mitigation are needed. These strategies rely on comprehensive hydrological and climatic models and the

corresponding data inputs.

**Table 1.** General characteristics of selected study regions

| Study Locations | Area (km²) | Average area elevation (m) | Number of stations | Precipitation (mm) | | | | |
|---|---|---|---|---|---|---|---|---|
| | | | | Northeast Monsoon (October to December) | Southwest Monsoon (June to September) | Winter Season (January to February) | Summer Season (March to May) | Annual |
| Coimbatore | 4732 | 427 | 26 | 343.8 | 686.8 | 19.6 | 164.8 | 1215.0 |
| Madurai | 3710 | 101 | 12 | 418.8 | 325.2 | 23.3 | 146.9 | 914.2 |
| Tiruchirappalli | 4509 | 74 | 15 | 394.2 | 276.6 | 18.1 | 96.7 | 785.6 |
| Tuticorin | 4745 | 27 to few metres | 16 | 427.7 | 64.7 | 41.4 | 113.9 | 647.7 |

**2.2 Data sets**

The reference data sets used for the evaluation of multiple data products are based on daily rainfall measured from

69 rain gauges. Station data for Coimbatore, Madurai, Tuticorin and Tiruchirappalli were provided by the Public Works Department (PWD), Government of Tamil Nadu, for the period 2003 to 2014. The daily data provided by the PWD were carefully and extensively checked for their quality, and data for 2005 and 2010 was eliminated due to the large number of missing inputs. Among the several global satellite and reanalysis-based precipitation datasets, those with the maximum time period coverage and the highest spatial resolution were chosen. Daily temporal scale precipitation products were selected

because ground station data was only available on a daily basis, even though several precipitation products were available at half-hourly and sub-daily temporal scales. We selected five satellite-based and three reanalysis-based rainfall estimates using the aforementioned standards (Table 2). The five satellite-based precipitation estimates fulfilling are the TRMM 3B42(Climate Data Guide, 2024), PERSIANN CDR (Ashouri et al., 2015), CMORPH (Joyce et al., 2004), GPM (Hou et al., 2014) and MSWEP (Beck et al., 2017). Also, the reanalysis products considered are MERRA-2(Gelaro et al., 2017), NCEP2 and ERA5-

Land. Reanalysis products like 20CRv2, CFSR and JRA-55 are not considered in this study due to their limited spatial and temporal resolution.

Launched in late 1997, the TRMM products are among the satellite precipitation products that have been used extensively because of their acceptable accuracy (Climate Data Guide, 2024; Sahoo et al., 2015). The mission uses 5 instruments: Precipitation Radar (PR), TRMM Microwave Imager (TMI), Visible Infrared Scanner (VIRS), Clouds and Earths

Radiant Energy System (CERES) and Lightning Imaging Sensor (LSI).  The TMI and PR are the main instruments used for



precipitation. These instruments are used in an algorithm that forms the TRMM Combined Instrument (TCI) calibration data set (TRMM 2B31) for the TRMM Multi-satellite Precipitation Analysis (TMPA), whose TMPA 3B43 monthly precipitation averages and TMPA 3B42 daily and sub-daily (3 hour) averages are probably the most relevant TRMM-related products for climate research. The product TRMM 3B42, which is available at 0.25° spatial resolution and daily temporal resolutions, is
used in this study.

PERSIANN-CDR is a satellite-based precipitation estimation product that provides more than three decades (from 1983 to present) of daily precipitation estimates at 0.25°×0.25° spatial resolution for the 60°S–60°N geographical extent. PERSIANN-CDR utilises the archive of infrared brightness temperature from the gridded satellite dataset (GridSat-B1) as the input of the PERSIANN algorithm (Hsu et al., 2000). The PERSIANN-CDR product was generated for each time step by
estimating precipitation for each GridSat-B1 Infrared Window (IRWIN) file using the basic PERSIANN algorithm, which used an Artificial Neural Network (ANN) to convert the input infrared data in degrees Kelvin into rain rate (RR) data in mm/hr. Each month of PERSIANN CDR estimates was then bias-corrected with monthly GPCP precipitation data and the final PERSIANN-CDR product results when those bias-correct precipitation estimates were accumulated daily. Then, the rainfall estimates of the PERSIANN CDR algorithm were bias corrected using the monthly Global Precipitation Climatology Project
(GPCP) version 2.3 product at 2.5°×2.5° spatial resolution.

CMORPH was generated on an 8kmx8km grid over the global domain (60°S-60°N) in 30-min intervals from January 1, 1998 to the present. The final product was constructed in two steps by integrating information from multiple satellites and in situ-based sources. The first step was to define integrated high-resolution global satellite precipitation estimates (the raw CMORPH). To this end, Level 2 precipitation rate retrievals from passive microwave (PMW) measurements aboard multiple
Low Earth Orbit (LEO) satellites were propagated from their respective observation times to the target analysis time along the cloud motion vectors derived from consecutive geostationary (GEO) infrared (IR) images. In the second step, bias in the raw CMORPH was removed through Probability Density Function (PDF) matching against the CPC daily gauge analysis over land (Xie et al., 2010) and through adjustment against the pentad GPCP merged analysis over ocean (Xie et al., 2003). The PMW data includes the data from TRMM Microwave Imager (TMI), Advanced Microwave Scanning Radiometer (AMSR), Micro-
Wave Radiation Imager (MWRI), Special Sensor Microwave Imager / Sounder (SSMIS), Special Sensor Microwave Imager (SSMI), Microwave Humidity Sounder (MHS) and Advanced Microwave Sounding Unit (AMSU). Three sets of CMORPH satellite precipitation estimates were included in the product package, providing global precipitation fields at a combination of time-space resolutions including a) 30-minute / 8km x 8km, b) hourly / 0.25° latitude/longitude, and c) daily / 0.25° latitude/longitude, respectively.

GPM Core Observatory satellite, launched on February 27, 2014, operates in low Earth orbit, carrying two instruments for measuring Earth's precipitation and serving as a calibration standard for other members of the GPM satellite constellation. It carries a passive multi-channel conical-scanning microwave radiometer called the GPM Microwave Imager (GMI) that can measure light to heavy precipitation and an active Dual-frequency Precipitation Radar (DPR) which can measure precipitation characteristics in the atmospheric column in three dimensions (Smith et al., 2007). It orbits in a non-sun-synchronous orbit



that takes it around Earth roughly 16 times per day, allowing it to sample precipitation at different times of the day. The Global Precipitation Measurement mission observes rain and snowfall worldwide every three hours, which contributes to the monitoring and forecasting of weather events such as droughts, floods and hurricanes, as well as scientific research on precipitation and climate change. GPM products are available to users at different levels of processing, with level 3 comprising geophysical parameters that have been spatially and/ or temporally resampled from Level 1 or Level 2 data. In this study, the

Level 3 IMERG Final Run product, which includes the research-quality gridded global multi-satellite precipitation estimates, was used.

       MSWEP V2 is the first fully global precipitation dataset with a 0.1° resolution derived by optimally merging a range of gauge, satellite, and reanalysis estimates (Beck et al., 2017). Its previous version was released in May 2016. MSWEP V2 merges the highest quality precipitation information from CPC morphing technique (CMORPH); Daily gauge data compiled

from GHCN-D, GSOP and other sources; European Centre for Medium Range Weather Forecasts (ECMWF) interim analysis (ERA-Interim); Global Precipitation Climatology Centre (GPCC) Full Data Reanalysis (FDR) V7 extended using First Guess (GPCC FDR), GridSat Infrared archive, Global Satellite Mapping of Precipitation (GSMaP) Moving Vector with Kalman (MVK), Japanese 55-year Reanalysis (JRA-55), TRMM Multi-satellite Precipitation Analysis (TMPA), WorldClim monthly climate datasets. It is a global precipitation dataset with 0.1° spatial resolution and has been available from 1979 to the present.

It takes advantage of the complementary strengths of gauge-, satellite-, and reanalysis-based data to provide reliable precipitation estimates over the entire globe. In this study, MSWEP version 2 with no gauge data is used.

       NCEP 2 was developed from NCEP1, which covered the period from 1948 onwards and is based on the T62 operational NCEP model in which the physical parameterisation was run on 192×94 Gaussian grid (1.92°×1.875°) (Kalnay et al., 2018). The model dynamics and physics follow a convective parameterisation scheme (Pan and Wu, 1995), which showed

a better and more realistic prediction of precipitation climatology. NCEP2 reanalysis (Kanamitsu et al., 2002) was run at the same resolution (T62) as NCEP1, covering the period from 1979 onwards. This product is free from many known biases and errors, which have been identified in the NCEP1. An important improvement was the updated precipitation parameterisations and more realistic cloud-top cooling. A simple assimilation scheme of rainfall has been employed to update soil moisture. The products are available in a 1.875°×1.875° grid with a 6-hour temporal resolution.

MERRA2 is a NASA atmospheric reanalysis that began in 1980. MERRA product was developed based on a version of the GEOS-5 atmospheric data assimilation system that was frozen in 2008 (Bosilovich et al., 2015). It was used to drive stand-alone reanalysis of the land surface (MERRA-Land) and atmospheric aerosols (MERRAero). MERRA2 was introduced to replace the original MERRA dataset because of the advances made in the assimilation system that enable the assimilation of modern hyperspectral radiance and microwave observations, along with GPS-Radio Occultation datasets. The current

version's advantages over the previous one include enhanced use of satellite observations, assimilation of aerosol information (MERRAero), NASA's ozone profile observations and advances in both the GEOS model and the GSI assimilation system (Cullather et al., 2014; Wu et al., 2002). MERRA2 grid has 576 points in the longitudinal direction and 361 points in the



latitudinal direction, corresponding to a resolution of 0.625°×0.5°. It is the first long-term global reanalysis to assimilate space-based observations of aerosols and represent their interactions with other physical processes in the climate system.

ERA5-Land describes the evolution of the water and energy cycles over land in a consistent manner over the production period. It covers the period from January 1950 to 2-3 months before the present. ERA5 was produced using 4D-Var data assimilation and model forecasts in CY41R2 of the ECMWF Integrated Forecast System (IFS), with 137 hybrid sigma/pressure (model) levels in the vertical and the top level at 0.01 hPa. ERA5-Land is a replay of the land component of the ERA5 climate reanalysis, forced by meteorological fields from ERA5. H-TESSEL is the land surface model that is the

basis of ERA5-Land.  The data is available on a regular latitude/longitude grid of 0.1° × 0.1°. In particular, ERA5-Land runs at enhanced resolution (9 km vs 31 km in ERA5). The temporal frequency of the output is hourly, and the fields are masked for all oceans, making them lighter to handle (Muñoz-Sabater et al., 2021).

**Table 2**. An overview of precipitation products used in the study

| No | Precipitation Product | Concept | Primary band/model | Station Data | Spatial* | Temp# |
|----|----|----|----|----|----|----|
| 1 | CMORPH | Data-driven | PMW – 10.7 to 190 GHz | CPC Gauge data (land), GPCP (Ocean) | 0.25° | Daily |
| 2 | IMERG-GPM Final Run | Data-driven | Dual Frequency Precipitation Radar - 35.5 GHz & 13.6 GHz; GPM Microwave Imager - 10 GHz to 183 GHz | GPCC gauge analysis | 0.1° | Daily |
| 3 | MSWEP (Past no gauge) | Data-driven | Weighted mean average merge | - | 0.1° | Daily |
| 4 | PERSIANN CDR | Data-driven | GridSat-B1 IR window - 11μm | GPCP monthly data | 0.25° | Daily |
| 5 | TRMM 3B42 | Data-driven | Precipitation Radar & TRMM Microwave Imager | GPCC gauge | 0.25° | Daily |
| 6 | ERA5-Land | Model-based | H-TESSEL | Gridded dataset based on station time series from ECA&D, US Climate Reference Network | 0.1° | Hourly |
| 7 | MERRA2 | Model-based | GEOS-5.12.4 system | Sea surface observations from International Comprehensive Atmosphere-Ocean Data set (ICOADS); Land surface observation from National Centers for Environmental Predictions (NCEP) | 0.5°×0.63° | Daily |
| 8 | NCEP2 | Model-based | T62 operational NCEP model | GPCC gauge | 1.875°×1.875° | Daily |

*Spatial resolution given in latitude × longitude, #Temporal resolution



## 3 Methodology

### 3.1 Study Regions and Ground Stations

The evaluation of multiple daily, monthly and yearly precipitation products was conducted for the selected ground stations of Coimbatore, Madurai, Tiruchirappalli and Tuticorin. The points in Fig. 1 are the geographical location of the available ground station data. In most regions of India, the density and availability of field-based meteorological stations are limited, and their accessibility is restricted for many reasons. Hence, the selection of ground station data was based on the availability, quality and density of field-based meteorological stations from 2003 to 2014. For quality reasons, the years 2005 and 2010 were excluded from the present study. Finally, datasets used for validation included 26 stations in Coimbatore, 12 stations in Madurai, 16 in Tuticorin and 15 in Tiruchirappalli.

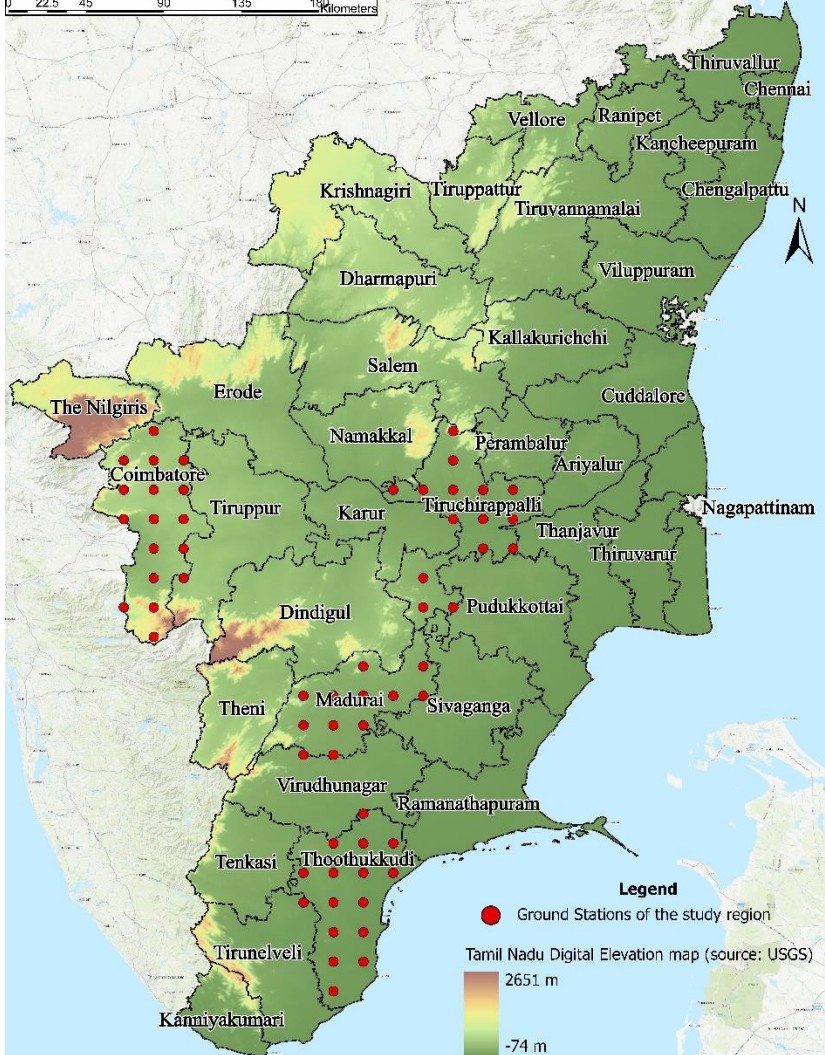

**Figure 1.** Map of Tamil Nadu and study regions (Coimbatore, Madurai, Tiruchirappalli, Tuticorin) with ground stations



## 3.2 Comparison of ground data with satellite and observational reanalysis-based data

The most commonly used method to compare ground observations with other data products, such as satellite-based precipitation estimates and reanalysis products, is point (station) to pixel comparison. It can be challenging to acquire
reasonable agreement when comparing daily rainfall, particularly in a very complex topography, on a point-to-pixel basis (Gebrechorkos et al., 2018). Therefore, in this study, we followed the most common practice of interpolating the point-based rain gauge data to 0.1° spatial resolution using the linear method, as it was relatively easy to implement and a popular application in evaluation studies. The interpolated 0.1° station dataset was used as ground truth to evaluate all the other precipitation products. Using the linear method, the satellite and reanalysis rainfall products with the coarse spatial resolution
were downscaled to 0.1°. The precipitation products were downscaled to higher resolution to capture spatial rainfall variability, which will be better exhibited at 0.1° as compared to their original coarse resolution. Also, we evaluated the performance of selected precipitation products at three temporal resolutions (daily, monthly and yearly). During the evaluation, we only considered the grids that contained at least one rain gauge. For each temporal resolution, evaluations were conducted at two spatial scales: the grid scale and the district scale. For district scale evaluations, the gridded precipitation (either interpolated
from station data or gridded precipitation products) from all grids containing at least one rain gauge was averaged to represent the precipitation distribution.

Evaluations were primarily based on the widely used statistical indices, such as the Pearson correlation coefficient (CC), Mean Absolute Error (MAE), Root Mean Square Error (RMSE), Bias (systematic error) Relative Bias ($R_{bias}$) and Index of Agreement (IA) were used. CC (Equation 1) is applied to evaluate the agreement of individual precipitation products (P) to
station data (O). A value of 1 shows a perfect fit between the products and station data.

$$\text{Correlation Coefficient} = \frac{\sum_{i=1}^{N}(P_i - \bar{P}).(O_i - \bar{O})}{\sqrt{\sum_{i=1}^{N}(P_i - \bar{P})^2}\ \sqrt{\sum_{i=1}^{N}(O_i - \bar{O})^2}} \qquad \text{(Equation 1)}$$

Each product's average differences and systemic bias are given as Bias (Equation 2) and $R_{bias}$ (Equation 3). Bias and $R_{bias}$ can be positive or negative depending on whether the product is underestimated or overestimated.

$$\text{Bias (Systematic error)} = \frac{\sum(P_i - O_i)}{N} \qquad \text{(Equation 2)}$$

$$R_{bias} = \frac{\sum_{i=1}^{N}(P_i - O_i)}{\sum_{i=1}^{N} O_i} \times 100 \qquad \text{(Equation 3)}$$

The MAE and RMSE (Equations 4 and 5) are well-known and accepted indicators of goodness of fit, which shows the difference between ground observations and precipitation products under study (Legates and McCabe Jr, 1999).

$$\text{MAE} = \frac{\sum_{i=1}^{N}|O_i - P_i|}{N} \qquad \text{(Equation 4)}$$

$$\text{RMSE} = \sqrt{\frac{\sum_{i=1}^{N}(O_i - P_i)^2}{N}} \qquad \text{(Equation 5)}$$



The IA (Willmott, 1981) is another widely used indicator of goodness of fit between station data and different products. Equation 6 describes the amount of precipitation product that is error-free as compared to the station data.

$$IA = \frac{\sum(P_i - O_i)^2}{\sum(|P - \bar{O}| + |O - \bar{O}|)}$$ (Equation 6)

Additional analyses were conducted at the district spatial scale, which includes Intensity-Duration-Frequency (IDF) curve analysis at a daily time scale, seasonal evaluation at a monthly timescale and Taylor plot-based multimodel comparison

at a yearly timescale. IDF curve analysis was done using the Gumbel distribution methodology. It is the most widely used distribution for IDF analysis owing to its suitability for modeling maxima. Frequency precipitation (PT in mm) for each duration with a specific return period T (in year) is given by the following equation:

$$PT = P_{ave} + KS$$ (Equation 7)

Where K is the Gumbel frequency factor, $P_{ave}$ is the average of the maximum precipitation corresponding to a specific duration,

and S is the standard deviation of the precipitation data.

Gumbel frequency factor, $K = \frac{\sqrt{6}}{\Pi}[0.5772 + \ln[\ln[\frac{T}{T-1}]]]$ (Equation 8)

Standard deviation of precipitation data $= \frac{1}{n-1}\sum_{i=1}^{n}(P_i - P_{ave})^2]^{0.5}$ (Equation 9)

Then the Precipitation Intensity, $I_t$ (mm/hour) for return period T is obtained from $I_t = \frac{P_t}{T_d}$ (Equation 10)

Where $T_d$ is the duration in hours

In the monthly timescale, Percent Mean Absolute Error (%MAE) (Equation 7) was computed as the ratio of MAE to the mean precipitation from corresponding station data. Hence, considering the precipitation's monthly variation, it enables the precipitation products' intercomparison (Duan et al., 2016).

$$\%MAE = \frac{Monthly\ MAE\ of\ the\ precipitation\ product}{Mean\ Precipitation\ from\ the\ Station\ data}$$ (Equation 11)

In addition to the above statistical methods, the Taylor diagram (Taylor, 2001) was used to summarize the statistical

relationship between ground station data and precipitation products at yearly timescale evaluations. This diagram explains the relationship between the two fields by correlation coefficient (R), standard deviation (σ) and centred root mean square (RMS) difference (E'). A single point on the diagram displays three statistical values (R, σ). Percent Bias, either in a positive or negative way, is indicated by the upward or downward notation of the product.

$$E'^2 = \sigma_f^2 + \sigma_r^2 - 2\sigma_f\sigma_r R$$ (Equation 12)

Where $\sigma_f^2$ and $\sigma_r^2$ are the variances of the satellite / reanalysis and station data, and R is the correlation coefficient between the two products. The distance between the reference point (station data) is given in the diagram as the centred RMS difference of the two fields (Equation 9). A satellite/reanalysis product with no error would show a perfect correlation to the station data.

$$E'^2 = \frac{1}{N}\sum_{n=1}^{N}[(f_n - \bar{f}) - (r_n - \bar{r})]^2$$ (Equation 13)

Where f is the model (e.g. satellite or reanalysis) data, and r is the reference (station data) data, whereas $\sigma_f$ and $\sigma_r$ are the

standard deviations of the model and reference fields (Equation 10 and Equation 11).



$$\sigma_f = \sqrt{\frac{1}{N} \sum_{n=1}^{N} (f_n - \bar{f})^2} \qquad \text{(Equation 14)}$$

$$\sigma_r = \sqrt{\frac{1}{N} \sum_{n=1}^{N} (r_n - \bar{r})^2} \qquad \text{(Equation 15)}$$

## 4 Results

### 4.1 Evaluation of precipitation products at the daily time scale

**4.1.1. Grid-Scale**

The statistical metrics for daily precipitation are summarized in Tables 3 and 4. For the study region Coimbatore at grid scale, ERA5-Land had the highest correlation value (CC = 0.366), followed by MSWEP. The lowest CC value of 0.052 was observed for the NCEP2 product. All the products overestimated the rainfall with a Bias and $R_{bias}$ value ranging from 0.197 to 18.293 and 0.087 to 8.036, respectively. MSWEP recorded the lowest RMSE, indicating the product's fit with the

station data compared to the other products. Interestingly, ERA5-Land was observed to have both the lowest MAE and highest IA. In Madurai, correlation trends were similar to those in Coimbatore. ERA5-Land, followed by MSWEP, had the highest correlation coefficient, while NCEP2 had the lowest correlation. ERA5-Land was observed to have the lowest MAE and RMSE values of 3.859 mm/day and 7.511 mm/day. Five precipitation products underestimated the rainfall with Bias and $R_{bias}$ values ranging from -2.039 to -1.123 and -0.456 to -0.253, respectively. On the other hand, NCEP2 and MERRA2 overestimated

precipitation, with the former having comparatively higher values of Bias and $R_{bias}$ (10.875 and 2.434). Regarding IA, ERA5-Land and NCEP2 recorded the highest and lowest values.

In Tiruchirappalli, correlation and IA had similar trends with respect to Coimbatore and Madurai. Table 4 shows that the correlation value ranged from 0.096 (NCEP2) to 0.371 (ERA5-Land). In terms of error, MERRA2 was observed to have the highest MAE (11.186 mm/day) and RMSE (33.08 mm/day). On the other hand, ERA5-Land had the lowest MAE (3.237

mm/day), and MSWEP had the lowest RMSE (7.229 mm/day). Five products underestimated the precipitation, with Bias and $R_{bias}$ ranging from -1.187 to -0.549 and -0.342 to -0.158, respectively. Furthermore, NCEP2 and MERRA2 overestimated the precipitation with values ranging above zero. ERA5-Land, followed by PERSIANN CDR, had the highest IA, while MERRA2 had the lowest value. In Tuticorin, MSWEP had the highest correlation (CC = 0.456), while NCEP2 had the lowest (CC = 0.117). MAE values ranged from 3.225 mm/day to 11.888 mm/day, with the lowest recorded by MSWEP and the highest

recorded by MERRA2. Similar trends were observed for RMSE, with values ranging from 6.623 to 33.268 mm/day. CMORPH, GPM, MSWEP, PERSIANN CDR, TRMM and ERA5-Land had underestimated the precipitation with negative Bias and $R_{bias}$ values. MERRA2 and NCEP2 overestimated rainfall with positive Bias and $R_{bias}$ values.

The performance of IDF curves generated using remote sensing and reanalysis-based precipitation products was evaluated by comparing them with the estimated IDF curves using station data. Rainfall intensity decreased with duration and

increased from lower to higher return periods. In the Coimbatore district, MSWEP and PERSIANN CDR had the closest



precipitation intensity estimation with respect to the station data during different time periods, as shown in Fig. 2a-c. ERA5-Land highly underestimated precipitation intensity, while GPM, CMORPH, TRMM, NCEP2 and MERRA2 overestimated. For the Madurai district, PERSIANN CDR, ERA5-Land and MSWEP underestimated the precipitation intensity, while GPM produced the closest estimate. From Fig. 2d-f, it is also observed that TRMM, CMORPH, NCEP2 and MERRA2 overestimated

the precipitation intensity. In Tiruchirappalli district, all the precipitation products overestimated the precipitation intensity in comparison to the station data (Fig. 2g-i). PERSIANN CDR produced the closest precipitation intensity estimation with slight overestimation. In the Tuticorin district, PERSIANN CDR underestimated the precipitation intensity, and ERA5-Land produced the closest estimate with respect to the station data. The remaining precipitation product overestimated the intensity at different time periods, with the highest being produced by MERRA2 (Fig. 2j-l).

The range of precipitation frequency at continuous intensity intervals (1-2, 2-5, 5-10, 10-20, 20-50 and >50 mm/day) was evaluated for the eight gridded precipitation products against the station data (Fig. S1). Since the frequency class 0–1 mm/day accounts for more than 60% of the daily grid-scale precipitation, it was investigated separately using Fig. S1. At the grid scale in Coimbatore, more than 65% of daily precipitation from the interpolated rain gauge falls in the intensity of 0-1 mm/day. At this intensity range, GPM (66%) had very close interpolation to the station data, followed by ERA5-Land (61%).

From the occurrence frequency of precipitation at a different intensity, as shown in Fig. S2a, CMORPH, GPM, and TRMM products followed the station data trend, underestimating the precipitation in the 1 to 20 mm/day range and thereafter overestimating it.

       For Madurai district, at grid scale, the frequency plot of 0-1 mm/day is shown in Fig. S1b. PERSIANN CDR product had a close estimate with the station data in a range of 0 to 1 mm/day. By observing the precipitation occurrence frequency at higher ranges (i.e. from 5 to 50 mm/day), CMORPH, GPM, TRMM, and ERA5-Land had produced underestimated values

consistently, as signified by the negative Bias value in Table 3. ERA5-Land and MSWEP were observed to have a similar trend as the gridded station data from 1-10 mm/day; thereafter, it diverged. Although NCEP2 had overestimated in the 2 to 10 mm/day range, it was found to capture gridded station data frequency at remaining intensity levels. Among the products, MERRA2 had the highest overestimation frequency percentage, which is highly substantiated by the positive Bias and $R_{bias}$

values in Table 3. In Tiruchirappalli, at grid scale, nearly 60 % of the interpolated ground station precipitation falls in the range of 0 to 1 mm/day (Fig. S1c). In frequency range from 1 mm/day to 50 mm/day, MSWEP was observed to have a close relation with the station data from 1 to 5 mm/ day, NCEP2 followed by PERSIANN CDR from 5 to 20 mm/ day, TRMM followed by GPM from 20 to 50 mm/day and ERA5-Land at more than 50 mm/ day range, respectively (Fig. S2c). Further, Fig. 2d shows the grid scale precipitation data of Tuticorin at 0 to 1 mm/day intensity range. It is observed that more than 55%

of the interpolated ground station precipitation falls in the 0 to 1 mm/day range, which was closely captured by NCEP2 and





PERSIANN CDR. In the precipitation range of 0-1 mm/day, NCEP2, followed by PERSIANN CDR, had estimated closer frequency percentages with respect to the interpolated ground station data. Further, MSWEP is followed by ERA5-Land at 1 to 2 mm/day, NCEP2 followed by MSWEP at 2 to 5 mm/day, NCEP2 at 5 to 20 mm/day, TRMM at 20 to 50 mm/day and PERSIANN CDR at precipitation range above 50 mm/day.


### 4.1.2 District-Scale

The statistical evaluations perform better at the district scale and typically reflect the grid scale trend. In Coimbatore, ERA5-Land recorded the lowest RMSE with 0.614 mm/day as compared to MSWEP at the grid scale. Also, PERSIANN CDR was observed to have the lowest Bias, $R_{bias}$ and highest IA. In Madurai, the statistical metrics at the district level followed the grid scale trend. In Tiruchirappalli, ERA5-Land recorded the lowest RMSE with 6.663 mm/day, and other metrics followed the grid scale trend. In Tuticorin, PERSIANN CDR was observed to have the highest correlation coefficient of 0.466 and the lowest RMSE of 6.025 mm/day.



**Table 3.** Daily precipitation characteristics of Coimbatore and Madurai

| Scale | Product | Coimbatore | | | | | | Madurai | | | | | |
|---|---|---|---|---|---|---|---|---|---|---|---|---|---|
| | | CC | MAE | RMSE | Bias | RB | IA | CC | MAE | RMSE | Bias | RB | IA |
| Grid | CMORPH | 0.19 | 3.62 | 8.92 | 0.58 | 0.26 | 0.36 | 0.24 | 5.21 | 10.92 | -1.12 | -0.25 | 0.47 |
| | GPM | 0.21 | 3.87 | 9.00 | 1.00 | 0.44 | 0.38 | 0.29 | 4.67 | 9.20 | -1.60 | -0.36 | 0.53 |
| | MSWEP | 0.33 | 2.96 | 6.07 | 0.39 | 0.16 | 0.55 | 0.41 | 4.09 | 7.87 | -1.85 | -0.41 | 0.61 |
| | PERSIANN CDR | 0.20 | 4.25 | 7.72 | 0.37 | 0.16 | 0.56 | 0.35 | 4.52 | 8.00 | -1.95 | -0.43 | 0.62 |
| | TRMM | 0.21 | 3.49 | 8.76 | 0.20 | 0.09 | 0.30 | 0.26 | 5.09 | 10.13 | -1.26 | -0.28 | 0.50 |
| | ERA5-Land | 0.37 | 2.86 | 6.32 | 0.35 | 0.15 | 0.57 | 0.45 | 3.86 | 7.51 | -2.04 | -0.46 | 0.63 |
| | MERRA2 | 0.24 | 18.95 | 41.36 | 18.29 | 8.04 | 0.13 | 0.31 | 13.0 | 35.74 | 10.87 | 2.43 | 0.25 |
| | NCEP2 | 0.05 | 5.40 | 15.77 | 2.59 | 1.14 | 0.13 | 0.07 | 6.44 | 14.00 | 0.45 | 0.10 | 0.28 |
| District | CMORPH | 0.23 | 3.39 | 8.09 | 0.58 | 0.26 | 0.39 | 0.25 | 5.12 | 10.56 | -1.12 | -0.25 | 0.48 |
| | GPM | 0.25 | 3.55 | 8.03 | 1.00 | 0.44 | 0.41 | 0.31 | 4.56 | 8.81 | -1.60 | -0.36 | 0.55 |
| | MSWEP | 0.38 | 2.70 | 5.37 | 0.39 | 0.16 | 0.59 | 0.43 | 4.00 | 7.59 | -1.85 | -0.41 | 0.63 |
| | PERSIANN CDR | 0.23 | 4.04 | 7.15 | -0.17 | -0.15 | 0.59 | 0.36 | 4.44 | 7.80 | -1.95 | -0.43 | 0.64 |
| | TRMM | 0.23 | 0.17 | 1.87 | 1.43 | 0.63 | 0.37 | 0.27 | 4.99 | 9.78 | -1.26 | -0.28 | 0.52 |
| | ERA5-Land | 0.39 | 0.06 | 0.61 | -0.73 | -0.45 | 0.59 | 0.48 | 3.79 | 7.24 | -2.04 | -0.46 | 0.65 |
| | MERRA2 | 0.29 | 18.7 | 39.9 | 18.29 | 8.04 | 0.14 | 0.32 | 12.9 | 35.41 | 10.87 | 2.43 | 0.25 |
| | NCEP2 | 0.06 | 5.22 | 15.51 | 2.59 | 1.14 | 0.13 | 0.07 | 6.40 | 13.91 | 0.45 | 0.10 | 0.29 |


**Table 4.** Daily precipitation characteristics of Tiruchirappalli and Tuticorin

| Scale | Product | Tiruchirappalli | | | | | | Tuticorin | | | | | |
|---|---|---|---|---|---|---|---|---|---|---|---|---|---|
| | | CC | MAE | RMSE | Bias | RB | IA | CC | MAE | RMSE | Bias | RB | IA |
| Grid | CMORPH | 0.20 | 4.19 | 9.46 | -0.97 | -0.28 | 0.42 | 0.33 | 3.88 | 8.50 | -1.66 | -0.45 | 0.55 |
| | GPM | 0.26 | 4.19 | 9.53 | -0.55 | -0.16 | 0.47 | 0.38 | 3.99 | 7.94 | -1.26 | -0.34 | 0.60 |
| | MSWEP | 0.33 | 3.45 | 7.23 | -1.15 | -0.33 | 0.56 | 0.46 | 3.23 | 6.62 | -0.99 | -0.29 | 0.66 |
| | PERSIANN CDR | 0.33 | 3.91 | 7.32 | -1.17 | -0.33 | 0.57 | 0.43 | 3.62 | 6.53 | -1.21 | -0.34 | 0.63 |
| | TRMM | 0.23 | 4.27 | 9.65 | -0.67 | -0.19 | 0.44 | 0.36 | 3.96 | 8.47 | -1.53 | -0.41 | 0.58 |
| | ERA5-Land | 0.37 | 3.24 | 6.92 | -1.19 | -0.34 | 0.59 | 0.39 | 3.57 | 7.23 | -1.43 | -0.39 | 0.61 |
| | MERRA2 | 0.21 | 11.19 | 33.08 | 9.38 | 2.70 | 0.18 | 0.30 | 11.89 | 33.27 | 10.49 | 2.83 | 0.23 |
| | NCEP2 | 0.10 | 5.50 | 13.33 | 1.06 | 0.31 | 0.26 | 0.12 | 6.16 | 13.59 | 1.48 | 0.40 | 0.30 |
| District | CMORPH | 0.21 | 4.11 | 9.16 | -0.97 | -0.28 | 0.43 | 0.37 | 3.69 | 7.75 | -1.66 | -0.45 | 0.59 |
| | GPM | 0.27 | 4.08 | 9.10 | -0.55 | -0.16 | 0.48 | 0.43 | 3.62 | 7.24 | -1.26 | -0.34 | 0.64 |
| | MSWEP | 0.35 | 3.36 | 6.94 | -1.15 | -0.33 | 0.57 | 0.46 | 3.19 | 6.53 | -0.99 | -0.29 | 0.66 |
| | PERSIANN CDR | 0.35 | 3.81 | 7.08 | -1.17 | -0.33 | 0.58 | 0.47 | 3.38 | 6.03 | -1.21 | -0.34 | 0.66 |
| | TRMM | 0.24 | 4.17 | 9.30 | -0.67 | -0.19 | 0.45 | 0.40 | 3.74 | 7.73 | -1.53 | -0.41 | 0.61 |
| | ERA5-Land | 0.39 | 3.15 | 6.66 | -1.19 | -0.34 | 0.60 | 0.46 | 3.28 | 6.37 | -1.43 | -0.39 | 0.66 |
| | MERRA2 | 0.22 | 11.09 | 32.84 | 9.38 | 2.70 | 0.17 | 0.33 | 11.62 | 32.24 | 10.49 | 2.83 | 0.23 |
| | NCEP2 | 0.10 | 5.43 | 13.21 | 1.06 | 0.31 | 0.26 | 0.12 | 6.00 | 13.36 | 1.48 | 0.40 | 0.29 |





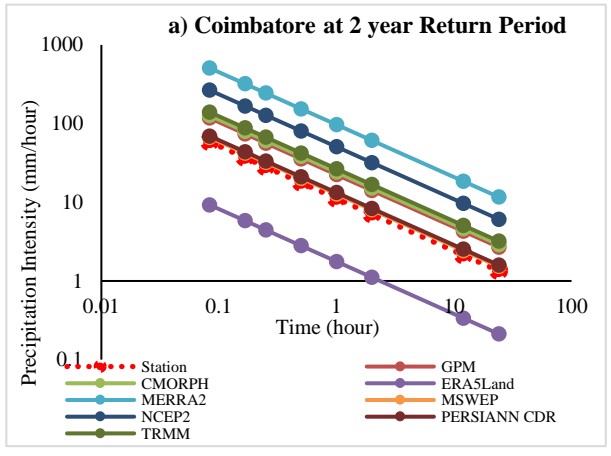
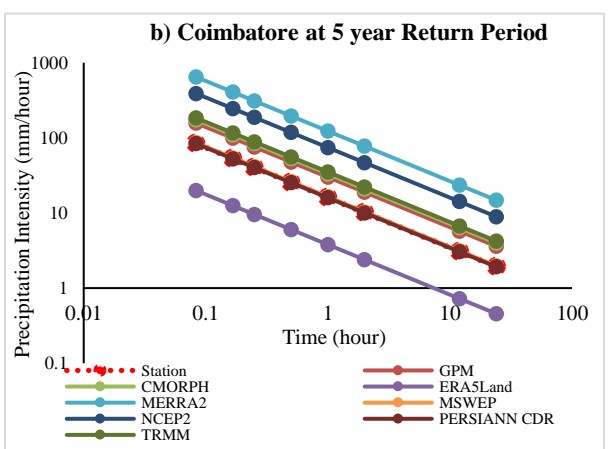

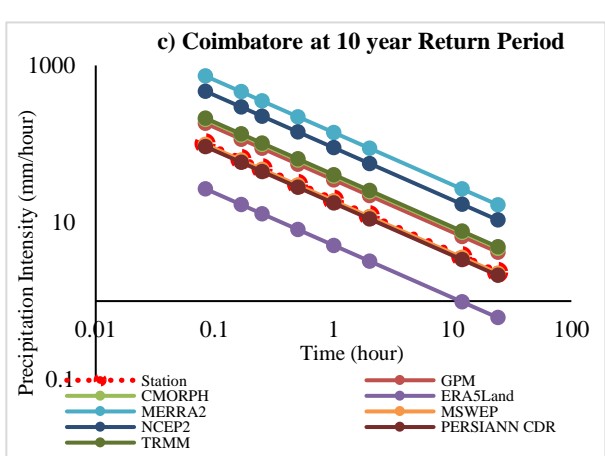
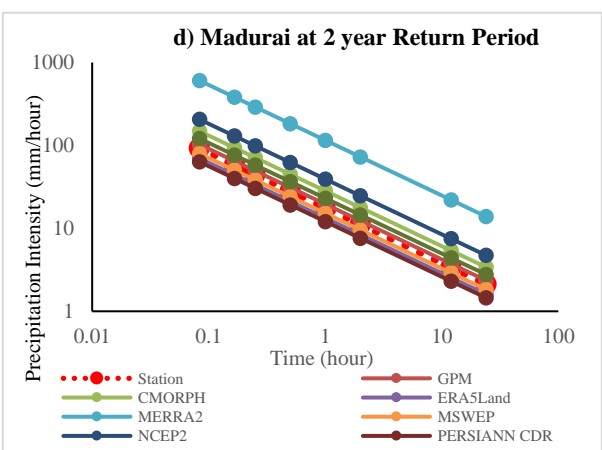

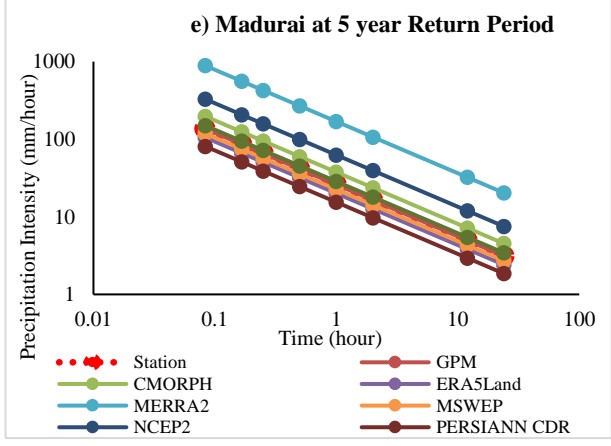
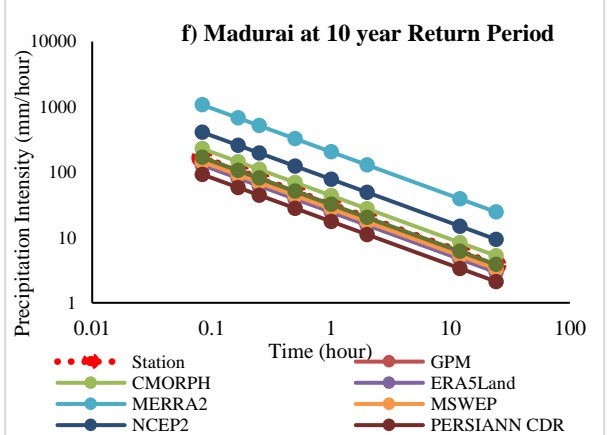





**Figure 2.** Intensity Duration Frequency (IDF) curves based on daily extreme (maximum) precipitation values
for the study regions at 2,5, 10 years return period



### 4.2 Evaluation of precipitation products at the monthly time scale

#### 4.2.1 Grid Scale

When comparing the estimations to the daily time scale, the monthly timeframe exhibited a considerably lower error and stronger correlations with the station data. This could be because statistical metrics take less variability into account when
analysing data with a lower temporal resolution. At the grid scale in Coimbatore, ERA5-Land had the highest correlation coefficient, followed by MSWEP (Table 5). The precipitation products NCEP2 and MERR2 were observed to have a correlation with the station data, indicating poor representation at this spatial scale. Further, ERA5-Land was observed to have the lowest MAE and RMSE, with values of 43.632 mm/month and 62.799 mm/month, respectively. All the products were observed to overestimate the grid scale monthly rainfall, with MERRA2 having the highest Bias and $R_{bias}$ values. On the other
hand, TRMM and ERA5-Land were observed to have the lowest Bias value and $R_{bias}$ values of 0.197 and 0.079, respectively. Further, MSWEP had the highest IA and NCEP2 had the lowest. In the Madurai region, the performance of the precipitation products at the grid was observed to be highly varied, as shown in Table 5. ERA5-Land recorded the highest correlation coefficient of 0.814, followed by MSWEP (CC = 0.8). TRMM was observed to record the lowest MAE and RMSE. Further, five products underestimated the precipitation with negative Bias and $R_{bias}$ values. NCEP2 and MERRA2 overestimated
precipitation at the monthly grid scale with positive Bias and $R_{bias}$ values. NCEP2 was observed to have the lowest Bias, whereas MERRA had the highest. Further, NCEP2 had the lowest IA and TRMM had the highest.

In Tiruchirappalli, ERA5-Land had the highest correlation coefficient (CC = 0.791), followed by PERSIANN CDR (r = 0.770). Also, as observed in Table 6, PERSIANN CDR recorded the lowest MAE, RMSE, Bias, $R_{bias}$, and IA. Five products underestimated monthly grid-scale precipitation at Tiruchirappalli, while two overestimated it. In Tuticorin, MSWEP was
observed to have the highest correlation coefficient (CC = 0.796) and IA (0.862). Further, MSWEP recorded the lowest MAE and RMSE of 47.07 mm/month and 66.35 mm/month. Among the products, PERSIANN CDR was found to have the minimum Bias and $R_{bias}$ values of -29.843 and -0.265, respectively (Table 6).





Table 5. Monthly precipitation characteristics of Coimbatore and Madurai

| Scale | Product | Coimbatore | | | | | | Madurai | | | | | |
|---|---|---|---|---|---|---|---|---|---|---|---|---|---|
| | | CC | MAE | RMSE | Bias | RB | IA | CC | MAE | RMSE | Bias | RB | IA |
| Grid | CMORPH | 0.54 | 53.79 | 72.09 | 17.60 | 0.26 | 0.72 | 0.16 | 66.47 | 91.80 | -34.17 | -0.25 | 0.79 |
| | GPM | 0.48 | 61.88 | 83.12 | 30.57 | 0.44 | 0.67 | 0.78 | 65.64 | 87.07 | -48.76 | -0.36 | 0.80 |
| | MSWEP | 0.59 | 45.99 | 63.01 | 8.71 | 0.12 | 0.83 | 0.80 | 67.68 | 91.04 | -58.34 | -0.43 | 0.79 |
| | PERSIANN CDR | 0.34 | 84.50 | 113.09 | 54.51 | 0.79 | 0.54 | 0.69 | 68.06 | 87.18 | -25.84 | -0.19 | 0.79 |
| | TRMM | 0.10 | 80.84 | 109.78 | 0.20 | 0.09 | 0.30 | 0.75 | 63.63 | 84.70 | -38.29 | -0.28 | 0.82 |
| | ERA5-Land | 0.62 | 43.63 | 62.80 | 5.70 | 0.08 | 0.78 | 0.81 | 69.98 | 93.25 | -62.09 | -0.46 | 0.78 |
| | MERRA2 | -0.15 | 68.78 | 96.82 | 556.73 | 8.04 | 0.16 | 0.71 | 333.84 | 459.54 | 330.96 | 2.43 | 0.42 |
| | NCEP2 | -0.08 | 121.22 | 225.95 | 78.79 | 1.14 | 0.13 | -0.02 | 131.16 | 190.56 | 13.65 | 0.10 | 0.30 |
| District | CMORPH | 0.63 | 47.69 | 60.63 | 17.60 | 0.26 | 0.77 | 0.69 | 65.44 | 89.22 | -34.17 | -0.25 | 0.79 |
| | GPM | 0.56 | 54.81 | 73.52 | 30.57 | 0.44 | 0.71 | 0.80 | 64.83 | 85.03 | -48.76 | -0.36 | 0.81 |
| | MSWEP | 0.69 | 38.43 | 50.66 | 8.71 | 0.12 | 0.83 | 0.81 | 67.00 | 89.25 | -58.34 | -0.43 | 0.80 |
| | PERSIANN CDR | 0.39 | 78.23 | 107.45 | 54.51 | 0.79 | 0.54 | 0.70 | 67.76 | 85.39 | -25.84 | -0.19 | 0.80 |
| | TRMM | -0.01 | 2.28 | 14.08 | 25.22 | 0.50 | 0.30 | 0.77 | 62.88 | 82.40 | -38.29 | -0.28 | 0.83 |
| | ERA5-Land | 0.72 | 35.84 | 50.35 | 5.70 | 0.08 | 0.84 | 0.83 | 69.45 | 91.38 | -62.09 | -0.46 | 0.78 |
| | MERRA2 | 0.56 | 556.6 | 705.76 | 556.73 | 8.04 | 0.15 | 0.72 | 333.68 | 457.92 | 330.96 | 2.43 | 0.42 |
| | NCEP2 | -0.09 | 117.6 | 223.0 | 78.79 | 1.14 | 0.12 | -0.02 | 130.84 | 189.86 | 13.67 | 0.10 | 0.30 |

Table 6. Monthly precipitation characteristics of Tiruchirappalli and Tuticorin

| Scale | Product | Tiruchirappalli | | | | | | Tuticorin | | | | | |
|---|---|---|---|---|---|---|---|---|---|---|---|---|---|
| | | CC | MAE | RMSE | Bias | RB | IA | CC | MAE | RMSE | Bias | RB | IA |
| Grid | CMORPH | 0.64 | 55.51 | 79.11 | -29.57 | -0.28 | 0.77 | 0.73 | 60.42 | 83.80 | -50.57 | -0.45 | 0.79 |
| | GPM | 0.71 | 51.08 | 72.25 | -16.67 | -0.16 | 0.83 | 0.77 | 55.31 | 75.04 | -38.41 | -0.34 | 0.83 |
| | MSWEP | 0.75 | 50.99 | 69.96 | -36.54 | -0.35 | 0.82 | 0.80 | 47.07 | 66.35 | -32.02 | -0.31 | 0.86 |
| | PERSIANN CDR | 0.77 | 43.51 | 61.65 | -7.32 | -0.07 | 0.87 | 0.73 | 50.59 | 69.23 | -29.84 | -0.26 | 0.83 |
| | TRMM | 0.69 | 51.99 | 74.32 | -20.43 | -0.19 | 0.82 | 0.77 | 57.74 | 77.44 | -46.41 | -0.41 | 0.82 |
| | ERA5-Land | 0.79 | 49.01 | 66.13 | -36.16 | -0.34 | 0.83 | 0.65 | 62.03 | 89.45 | -43.45 | -0.39 | 0.77 |
| | MERRA2 | 0.59 | 17383.95 | 416.93 | 285.37 | 2.70 | 0.34 | 0.64 | 319.79 | 448.33 | 319.31 | 2.83 | 0.37 |
| | NCEP2 | 0.17 | 103.07 | 160.10 | 32.27 | 0.31 | 0.40 | 0.11 | 114.70 | 169.54 | 45.12 | 0.40 | 0.38 |
| District | CMORPH | 0.65 | 53.66 | 76.60 | -29.57 | -0.28 | 0.78 | 0.80 | 57.18 | 75.62 | -50.57 | -0.45 | 0.82 |
| | GPM | 0.73 | 48.52 | 69.47 | -16.67 | -0.16 | 0.84 | 0.80 | 51.76 | 68.40 | -38.41 | -0.34 | 0.85 |
| | MSWEP | 0.77 | 49.13 | 67.24 | -36.54 | -0.35 | 0.83 | 0.80 | 46.72 | 65.58 | -32.02 | -0.31 | 0.86 |
| | PERSIANN CDR | 0.79 | 41.04 | 58.07 | -7.32 | -0.07 | 0.89 | 0.82 | 44.03 | 61.57 | -29.84 | -0.26 | 0.86 |
| | TRMM | 0.71 | 50.09 | 71.46 | -20.43 | -0.19 | 0.83 | 0.83 | 55.92 | 70.34 | -46.41 | -0.41 | 0.84 |
| | ERA5-Land | 0.81 | 47.53 | 63.27 | -36.16 | -0.34 | 0.84 | 0.76 | 55.56 | 75.15 | -43.45 | -0.39 | 0.82 |
| | MERRA2 | 0.60 | 287.05 | 415.75 | 285.37 | 2.70 | 0.34 | 0.68 | 319.43 | 443.23 | 319.31 | 2.83 | 0.37 |
| | NCEP2 | 0.18 | 102.69 | 158.98 | 32.27 | 0.31 | 0.40 | 0.11 | 113.03 | 167.21 | 45.12 | 0.40 | 0.37 |




### 4.2.2 District Scale

All the precipitation products overestimated rainfall at the district scale in Coimbatore, similar to their performance at the grid scale. ERA5-Land recorded the highest CC and IA in the Coimbatore district scale. Contrasting to grid-scale

performance, TRMM produced the lowest MAE (35.842 mm/month) and RMSE (50.348 mm/month), which is followed by ERA5-Land (Table 5). District scale performance of precipitation products at Madurai (Table 5) and Tiruchirappalli (Table 6) mostly depicted similar trends as observed in the grid scale. MAE trends at the district scale differed from those of grid-scale performance in Madurai. TRMM (62.883 mm/month) and MERRA2 (333.675 mm/month) recorded the lowest and highest MAE in Madurai. In Tuticorin, TRMM recorded the highest CC, followed by GPM (Table 6). In terms of error, PERSIANN

CDR was found to have the lowest MAE (44.033 mm/month) and RMSE (61.570 mm/month). The performance of Bias, $R_{bias}$, and IA followed a similar trend to that of the grid scale.

Further, we also investigated the seasonal trend by including the stacked plot of %MAE during monsoon (October to December) and non-monsoon months (January to November) of the study period. Figure 3a shows that ERA5-Land, followed by CMORPH and GPM, closely depicts the station's monsoon precipitation at Coimbatore. On the other hand, all the products

overestimated the precipitation for the non-monsoon month, with CMORPH having the closest fit. From Fig. 3b, monsoon months have a lower %MAE compared to the non-monsoon months. CMORPH, followed by PERSIANN CDR, recorded the lowest %MAE during the monsoon and non-monsoon months at Coimbatore. In Madurai, most of the products underestimated the monsoon month precipitation while MERRA2 overestimated it (Fig. 4a). During non-monsoon months, PERSIANN CDR had the closest depiction with the station precipitation data. Fig. 4b shows that ERA5-Land and MERRA2 had the lowest

monsoon month %MAE, respectively. On the other hand, CMORPH, followed by MSWEP, had the minimum %MAE during non-monsoon months at Madurai. In Tiruchirappalli, except MERRA2, all the products underestimated the monsoon month precipitation. Fig. 5a depicts that GPM only produced the closest value with respect to the station data. Further, CMORPH, followed by PERSIANN CDR, had the closest estimation during the non-monsoon months. ERA5-Land recorded the lowest %MAE during both periods, as shown in Fig. 5b. Like Madurai and Tiruchirappalli, monsoon month precipitation was

underestimated by most of the products in Tuticorin (Fig. 6a). MSWEP and PERSIANN CDR estimated the closest value with respect to the station data during the monsoon and non-monsoon months, respectively. Further, ERA5-Land estimated the lowest %MAE during the monsoon months and PERSIANN CDR during the non-monsoon months (Fig. 6b).




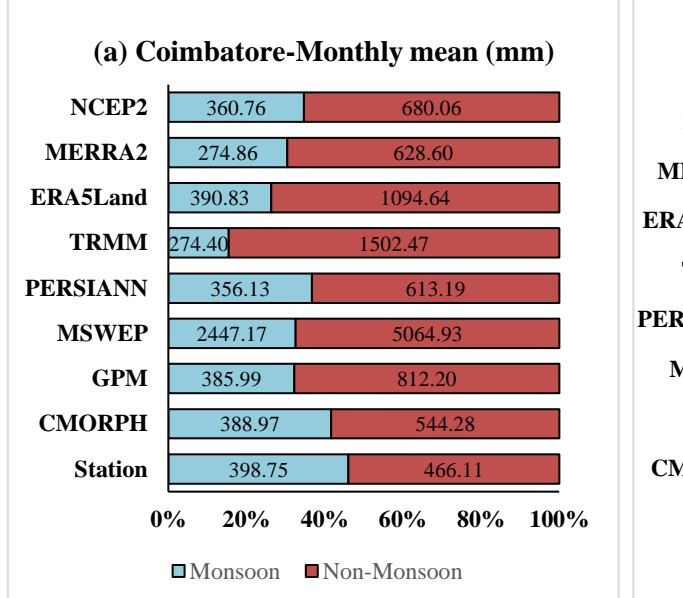
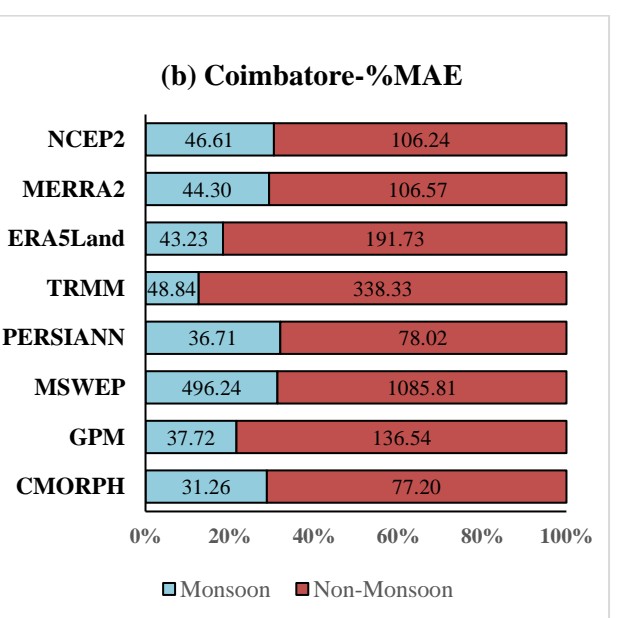

**Figure 3.** Monthly Mean (a) and %MAE (b) of different precipitation products with respect to Station data in Coimbatore

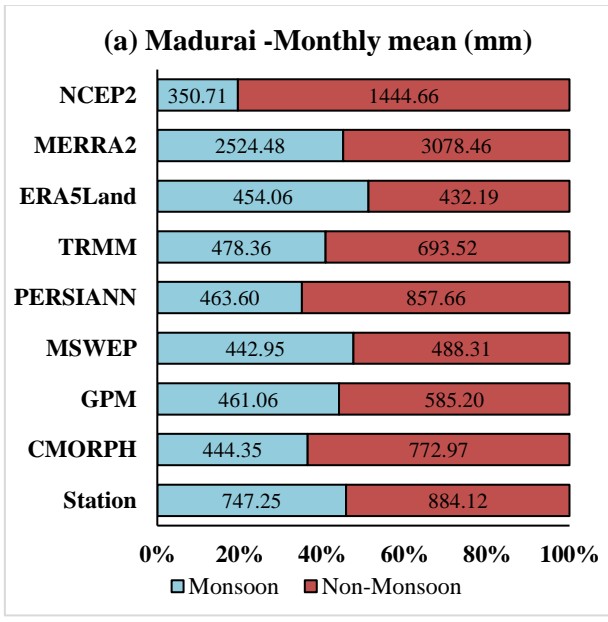
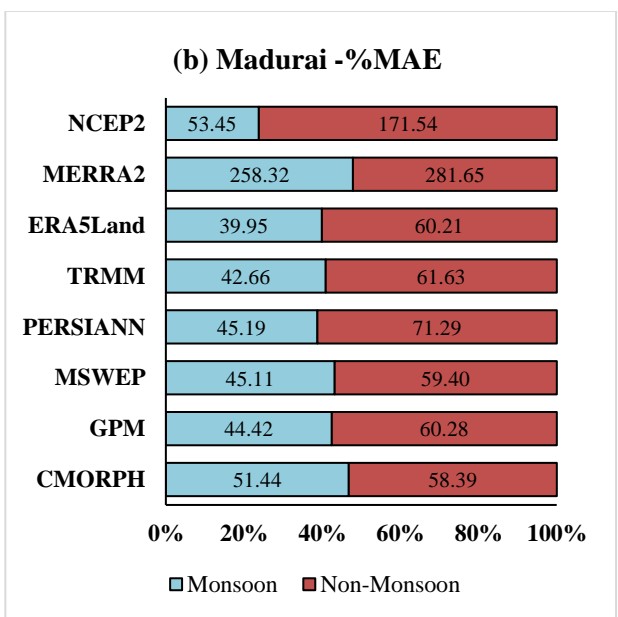

**Figure 4.** Monthly Mean (a) and %MAE (b) of different precipitation products with respect to Station data in Madurai





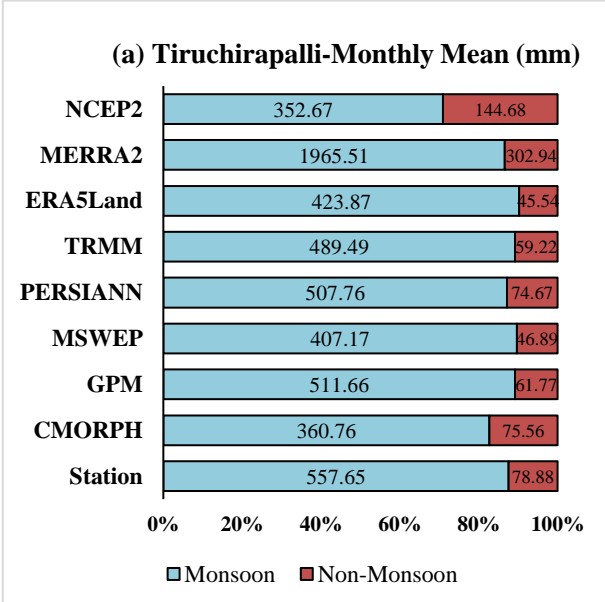
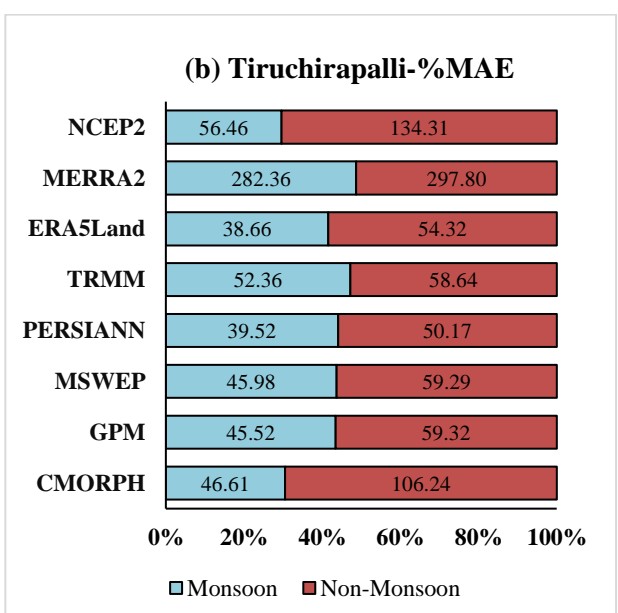

**Figure 5.** Monthly Mean (a) and %MAE (b) of different precipitation products with respect to Station data in Tiruchirappalli

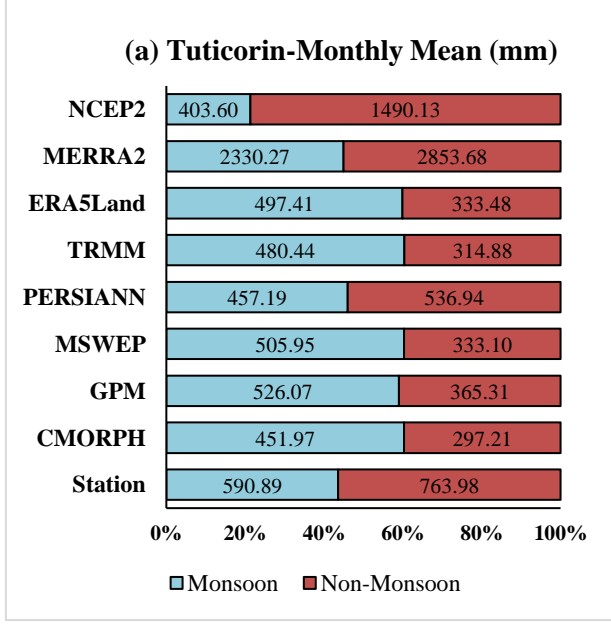
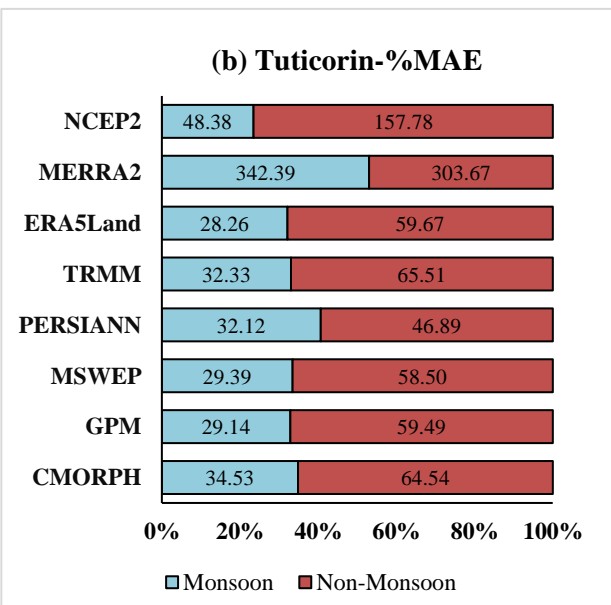

**Figure 6.** Monthly Mean (a) and %MAE (b) of different precipitation products with respect to Station data in Tuticorin



### 4.3 Evaluation of precipitation products at the annual time scale

**4.3.1 Grid Scale**

The grid-scale daily precipitation was summed as the annual precipitation and was statistically evaluated against the station data. PERSIANN CDR recorded the highest CC in Coimbatore, while ERA5-Land had the best metrics regarding MAE, RMSE and IA. All the products overestimated the precipitation, with the highest overestimation caused by ERA5-Land (Table 7). In Madurai and Tiruchirappalli (Table 8), PERSIANN CDR was found to have the lowest MAE, RMSE and highest IA.

On the other hand, ERA5-Land had the highest CC. Except for NCEP2 and MERRA2, all the products underestimated the precipitation. In Tuticorin, MSWEP was found to have the highest CC, IA and lowest RMSE (Table 8). PERSIANN CDR recorded the lowest MAE, Bias and $R_{bias}$.

The yearly precipitation was further evaluated with Taylor Plots. It enables the comparison of multiple models with reference to the station value in a single plot. In general, precipitation products show overestimation in Coimbatore and

underestimation in the remaining study regions, as substantiated by their Bias value in Tables 7 and 8. Also, NCEP2 and MERRA2 could not be included in the same plot as the other products due to their higher standard deviation values. By examining the scatter of the product's RMS at the grid spatial scale, PERSIANN CDR had the closest relationship to the station data in Coimbatore, ERA5-Land in Madurai and Tiruchirappalli, MSWEP in Tuticorin (Fig. 7). Similar trends were observed at the district scale, except in Tuticorin, where ERA5-Land lay closer to the station data (Fig. S3). From the combined Taylor

plot at the annual scale, it is observed that ERA5-Land performance in Tiruchirappalli is the closest to the origin. The RMS spread of GPM, ERA5-Land, and PERSIANN CDR are similar in Madurai. Partial Clustering of PERSIANN CDR with GPM is observed in Tiruchirappalli. Most of the precipitation product's RMS deviates from the observed data for the Coimbatore region (Fig. S4a). For the combined Taylor Plot in monsoon months, GPM of Madurai lies the closest to the station RMS, as shown in Figure S4b. Unlike annual and non-monsoon plots, NCEP2 was observed to be included in the monsoon Taylor plot.

Performance of precipitation products like TRMM, PERSIANN CDR of Madurai and Tuticorin is clustered around the similar RMS value. During non-monsoon months, ERA5-Land in Tiruchirappalli lies closer to the standard deviation of the station. The deviation of precipitation products from station standard deviation is greater in Coimbatore for non-monsoon months. Clustering of multiple precipitation products for Tuticorin is observed, indicating their similarity in performance (Fig. S4c).

Geospatial maps of the annual rainfall were plotted using Inverse Distance Weighing (IDW) of the grid-scale values

using ArcGIS Pro software. The accuracy of different precipitation products in terms of capturing the variability can be assessed through these plots. The station values from Coimbatore showed a large variability from south to north. The annual precipitation varied from 400 mm in the southern region to 1200 mm in the northern region. Also, a few pockets in the northern region received precipitation higher than 1200 mm. CMORPH and partially GPM represented the precipitation in the northern region. All the products overestimated the precipitation in the southern region, and only ERA5-Land produced the closest

approximation to the station data (Fig. 8). In Coimbatore, CMORPH and GPM closely depicted the




station precipitation pattern in the northern regions. On the other hand, ERA5-Land produced the closest estimate in the southern portion.

Madurai district mostly had an even distribution of precipitation within 1600-1800 mm (Fig. 9). In contrast to the results of statistical evaluation, NCEP2 had the closest estimates. Except for MERRA2, all the products have highly
underestimated the precipitation. Annual Precipitation in the Tiruchirappalli district falls in the range of 1100-1300 mm, with the eastern portion having 1300-1500 mm. Although PERSIANN CDR didn't capture the variability in the eastern region, it produced the closest estimate. Most of the precipitation products underestimated the precipitation and did not capture its regional variability (Fig. 10). In Tuticorin, a high precipitation variability is observed from the station data. The southern and partially eastern portions received precipitation in the 1400-1600 mm range, with some portions even exceeding 1600 mm
(Fig. 11). The remaining portions mostly had precipitation in the 1200-1400 mm range, and part of the western portion had 1000-1200 mm. Except for MERRA2 and NCEP2, all the other products underestimated the precipitation. Among the products, MSWEP and GPM produced the closest estimates compared to the station data.

**4.3.2 District Scale**

On a district scale, similar results were observed for Coimbatore, Madurai and Tiruchirappalli. In contrast to the grid scale performance, at the Tuticorin district scale, TRMM and ERA5-Land recorded the highest CC, while PERSIANN was found to have the lowest RMSE (Table 7 & 8).






**Table 7.** Yearly precipitation characteristics of Coimbatore and Madurai

| Scale | Product | | Coimbatore | | | | | Madurai | | | |
| | | CC | MAE | RMSE | RB | IA | CC | MAE | RMSE | RB | IA |
|---|---|---|---|---|---|---|---|---|---|---|---|
| Grid | CMORPH | 0.21 | 344.76 | 423.07 | 0.24 | 0.47 | 0.42 | 433.69 | 539.15 | -0.26 | 0.52 |
| | GPM | 0.12 | 454.90 | 536.78 | 0.44 | 0.44 | 0.64 | 585.09 | 644.06 | -0.36 | 0.50 |
| | MSWEP | 0.05 | 317.51 | 397.89 | 0.12 | 0.41 | 0.45 | 700.10 | 767.09 | -0.43 | 0.44 |
| | PERSIANN CDR | 0.39 | 670.97 | 731.66 | 0.79 | 0.44 | 0.61 | 330.26 | 416.82 | -0.19 | 0.60 |
| | TRMM | -0.61 | 860.22 | 939.29 | 0.09 | 0.07 | 0.48 | 462.26 | 560.14 | -0.28 | 0.54 |
| | ERA5-Land | 0.33 | 278.73 | 346.07 | 0.08 | 0.57 | 0.66 | 745.05 | 791.22 | -1.00 | 0.25 |
| | MERRA2 | -0.01 | 6680.71 | 6878.40 | 8.04 | 0.07 | 0.35 | 3971.54 | 4148.24 | 2.43 | 0.12 |
| | NCEP2 | -0.23 | 976.94 | 1145.97 | 1.14 | 0.26 | -0.75 | 471.97 | 609.66 | 0.10 | 0.07 |
| District | CMORPH | 0.42 | 9.69 | 56.92 | 0.24 | 0.53 | 0.44 | 61.43 | 200.58 | -0.26 | 0.52 |
| | GPM | 0.41 | 15.95 | 85.22 | 0.44 | 0.40 | 0.67 | 83.60 | 241.07 | -0.36 | 0.51 |
| | MSWEP | 0.43 | 12.50 | 53.03 | 0.12 | 0.58 | 0.47 | 100.03 | 287.87 | -0.43 | 0.45 |
| | PERSIANN CDR | 0.47 | 27.25 | 138.93 | 0.79 | 0.31 | 0.63 | 45.61 | 154.01 | -0.19 | 0.61 |
| | TRMM | 0.37 | 6.53 | 39.02 | 0.09 | 0.57 | 0.49 | 65.67 | 208.98 | -0.28 | 0.54 |
| | ERA5-Land | 0.69 | 9.72 | 41.64 | 0.08 | 0.80 | 0.71 | 106.46 | 296.84 | -0.46 | 0.46 |
| | MERRA2 | 0.30 | 278.36 | 1394.66 | 8.04 | 0.04 | 0.36 | 567.34 | 1565.42 | 2.43 | 0.12 |
| | NCEP2 | -0.48 | 39.40 | 225.68 | 1.14 | 0.12 | -0.77 | 66.67 | 228.17 | 0.10 | 0.07 |

**Table 8.** Yearly precipitation characteristics of Tiruchirappalli and Tuticorin

| Scale | Product | | Tiruchirappalli | | | | | Tuticorin | | | |
| | | CC | MAE | RMSE | RB | IA | CC | MAE | RMSE | RB | IA |
|---|---|---|---|---|---|---|---|---|---|---|---|
| Grid | CMORPH | 0.32 | 382.18 | 468.57 | -0.29 | 0.49 | 0.46 | 625.21 | 693.33 | -0.46 | 0.44 |
| | GPM | 0.68 | 230.68 | 283.43 | -0.16 | 0.72 | 0.71 | 461.92 | 517.85 | -0.34 | 0.55 |
| | MSWEP | 0.68 | 438.44 | 481.53 | -0.35 | 0.51 | 0.77 | 384.15 | 425.33 | -0.31 | 0.56 |
| | PERSIANN CDR | 0.68 | 174.31 | 217.84 | -0.07 | 0.80 | 0.48 | 379.93 | 463.14 | -0.26 | 0.50 |
| | TRMM | 0.60 | 266.01 | 333.92 | -0.19 | 0.65 | 0.60 | 558.37 | 617.91 | -0.41 | 0.48 |
| | ERA5-Land | 0.82 | 434.31 | 461.09 | -0.34 | 0.57 | 0.21 | 521.35 | 669.71 | -0.39 | 0.41 |
| | MERRA2 | 0.31 | 3424.42 | 3523.97 | 2.70 | 0.12 | 0.30 | 3831.64 | 3923.29 | 2.83 | 0.13 |
| | NCEP2 | -0.23 | 446.51 | 567.99 | 0.31 | 0.32 | -0.54 | 640.77 | 746.16 | 0.40 | 0.25 |
| District | CMORPH | 0.35 | 45.75 | 160.87 | -0.29 | 0.49 | 0.73 | 51.83 | 187.02 | -0.46 | 0.44 |
| | GPM | 0.74 | 25.64 | 92.69 | -0.16 | 0.74 | 0.88 | 38.42 | 138.19 | -0.34 | 0.54 |
| | MSWEP | 0.74 | 54.80 | 165.86 | -0.35 | 0.51 | 0.81 | 192.15 | 295.53 | -0.31 | 0.57 |
| | PERSIANN CDR | 0.79 | 17.66 | 62.87 | -0.07 | 0.85 | 0.85 | 29.85 | 113.78 | -0.26 | 0.56 |
| | TRMM | 0.67 | 30.63 | 110.92 | -0.19 | 0.67 | 0.90 | 46.43 | 164.62 | -0.41 | 0.49 |
| | ERA5-Land | 0.92 | 54.21 | 157.49 | -0.34 | 0.58 | 0.90 | 43.44 | 154.23 | -0.39 | 0.51 |
| | MERRA2 | 0.33 | 428.06 | 1245.12 | 2.70 | 0.11 | 0.44 | 319.30 | 1127.75 | 2.83 | 0.11 |
| | NCEP2 | -0.24 | 54.81 | 197.06 | 0.31 | 0.31 | -0.71 | 53.25 | 207.55 | 0.40 | 0.20 |







**Figure 7.** Taylor diagram depicting the agreement between station data and precipitation products at a yearly grid scale for (a) Coimbatore, (b) Madurai, (c) Tiruchirappalli and (d) Tuticorin





**Figure 8.** Mean annual precipitation for the Station data and different precipitation products at Coimbatore





**Figure 9.** Mean annual precipitation for the Station data and different precipitation products at Madurai





595



**Figure 10.** Mean annual precipitation for the Station data and different precipitation products at Tiruchirappalli



**Figure 11.** Mean annual precipitation for the Station data and different precipitation products at Tuticorin




**5 Discussion**

Tamil Nadu is an agriculturally significant state in India, contributing highly to the country's GDP. The state currently has 65% of its total area characterized as semi-arid, and more than 35% of the cropped area is rainfed (Government of Tamil Nadu, 2022). This climatic region's periodical moisture availability, when compounded by extreme variability, makes

irrigation, crop planning, and flood management more challenging. The four study regions belong to the state of Tamil Nadu, where the North-East monsoon is the principal source of precipitation. They are highly impacted by variable hydrometeorological hazards leading to human and financial loss. Accurate hydrological modeling is crucial for water resource planning, climate adaptation, and timely flood and drought forewarning (Mockler et al., 2016). For future hydrological security, modeling its changes and accompanying uncertainties is crucial (Loucks and Van Beek, 2017; Wheater and Gober, 2013;

Willems and De Lange, 2007). Under these circumstances, accurate hydrological modeling depends on precipitation data at the highest spatial and temporal resolution as a crucial input. Precipitation is characterized by a high variability in space, time and intensity (Prigent, 2010a). Historically, precipitation has been observed locally by rain gauges, which are sparsely distributed in a developing economy like India, leading to input uncertainty in hydrological studies (Renard, 1997; Wagener and Montanari, 2011). Hence, the present study aimed to evaluate the global precipitation products against the ground station

data at the grid scale (0.1°) and at the district scale. A grid-scale analysis will be crucial in determining if the precipitation product can accurately capture spatial variability. Previously, no analysis has been conducted at the district level. Given the highly variable nature of precipitation, both grid and district-level evaluations can provide valuable insights for scientific research and policymakers.

Comparing all the statistical metrics, ERA5-Land, followed by MSWEP, is the best precipitation product in

Coimbatore, Madurai and Tiruchirappalli. ERA5-Land is attributed as the best-performing product because of its higher CC combined with lower RMSE. Similar results have been reported for the Godavari River Basin in India by Reddy and Saravanan (2023). Further, PERSIANN CDR had lower MAE and RMSE in the three regions but substantially lower CC and IA. When the remaining precipitation products, GPM and TRMM—are considered, it is shown that GPM has a stronger correlation with less error than TRMM. The implementation of an advanced rainfall estimation algorithm might be the factor contributing to

the lower error of GPM (Beria et al., 2017; Pan et al., 2023). In accordance with our findings, Samykannu et al. (2021), also reported that GPM, TRMM and PERSIANN CDR produced a higher monthly correlation coefficient in comparison with the station data in Tamil Nadu. The study also reported that PERSIANN CDR produced the minimum RMSE in December. Of all the precipitation products examined in this study, only CMORPH exhibited good performance at Coimbatore during the monsoon season.

Tuticorin's precipitation product performance was different from the other research regions since it is a low-lying area with less precipitation (Tables 1 and 2). MSWEP performed the best in Tuticorin, followed by PERSIANN CDR. The MSWEP products consistently exhibited good performance for daily rainfall over the Indian subcontinent, as shown by a similar analysis



conducted by Nair and Indu (2017). This indicates that the product showed the best performance even with the exclusion of station data-based bias correction (Table 2).

MERRA2 and NCEP2 had the highest MAE and RMSE in all four study locations. They performed the worst by overestimating the precipitation in all studied regions, regardless of the spatial and temporal scale of evaluation. The higher grid dimension of the original product and the type of output data obtained from the atmospheric model in MERRA2 are responsible for the considerable inaccuracy. The impact of topographical elements, which have a vital influence on the precipitation process, can be overlooked by the enhanced spatial resolution (Singh and Singh, 2024). As a result, at smaller

scales, this product may depict precipitation occurrence inaccurately. (Table 2). Also, Gupta et al. (2024) reported that MERRA2 overestimated the summer month precipitation in Tamil Nadu and Andhra Pradesh.

The performance of precipitation products in representing extreme precipitation events varied from their overall performance. In all study regions except Tiruchirappalli, ERA5-Land underestimated the intensity of extreme rainfall for all three return periods (2, 5, and 10 years). Overall, MSWEP provided the most accurate estimates closest to the station data in

the majority of the study regions. The study by Yaswanth et al. (2023) also pointed out the limitations of satellite precipitation product-based modeling in capturing the peak discharge in the Adyar River basin. Furthermore, GPM was observed to yield more accurate estimates of heavy rainfall at the grid level over Peninsular India during the 2015 winter monsoon (Singh et al., 2018).

The precipitation products notably underestimated the monsoon month precipitation in all locations, except

Coimbatore. Table 2 indicates that most algorithms the products use combine high-temporal-frequency Visible/Infra-Red (V/IR) estimations with high-spatial-resolution Passive Microwave (MW) estimations. To address sampling issues caused by low orbits of current microwave sensors, passive microwave observations and infrared geostationary measurements are integrated. The continuous measurement provided by the infrared images effectively tracks system evolution, while the microwave rain retrieval serves as a reference (Sorooshian et al., 2000; Tapiador et al., 2004). The study areas, located in

tropical zones, are prone to convection-driven precipitation. In these regions, convection events can result in rain rates exceeding 50 mm/hour, lasting anywhere from less than 30 minutes to up to 6 hours (Prigent, 2010b). Despite the V/IR-MW integration being designed to capture light rainfall, it still leads to significant underestimations of precipitation in these areas. Coimbatore, however, being at a relatively higher altitude, is also subject to orographic precipitation (Table 1 and Fig. 1), leading to comparatively better performance of the precipitation products.

Moreover, based on the annual spatial plots, it is evident that the precipitation products were unable to accurately represent the spatial variability and generally underestimated the amount of precipitation. This aligns with similar observations in eastern China, where satellite-based precipitation products also underestimated precipitation and failed to capture the variability in the complex terrain and rainy areas of the south (Shaowei et al., 2022).

This study offers essential input for agro-hydrological modeling in the study region, addressing gaps left by previous

research. Beyond the standard statistical metrics, the study incorporates additional criteria for evaluating precipitation products. Intensity-Duration-Frequency (IDF) analysis is essential for selecting appropriate products for flood modeling in the context





of climate change and the associated rise in flood occurrences. Frequency plot analysis was performed to assess the differential capability of various products in capturing both low and high-intensity precipitation on a daily scale. Comparing the mean monthly values for monsoon and non-monsoon seasons will aid future studies in selecting the most suitable precipitation product for each season. Additionally, district-scale spatial plots on a yearly scale provide insights into the variability of precipitation across different parts of the districts, as recorded by ground station data, and evaluate the products' accuracy in capturing this variability.

In addition to directly integrating precipitation products for agro-hydrological modeling, many studies also make use of pre-compiled datasets. For example, in regions with limited ground stations, numerous agrometeorological studies rely on NASA Power for hydrometeorological modeling, yield predictions, and other analyses. As NASA Power is based on MERRA2 datasets, it is important to note that using it without bias correction can lead to significant errors, as the product tends to overestimate, as evident from the analysis of the study. Therefore, the findings of this study can help in selecting appropriate precipitation products and pre-compiled precipitation datasets, identifying biases at different time scales, and choosing the right bias correction methods. The main limitations of this study include the lack of access to station-based precipitation datasets after 2014 and the unavailability of hourly precipitation datasets for sub-daily analysis. Future studies could also examine the impact of different bias correction methods on enhancing the performance of various precipitation products.

## 6 Conclusion

In this study, the performance of multiple satellite-based (CMORPH, GPM, MSWEP, PERSIANN CDR and TRMM) and reanalysis-based (ERA5-Land, MERRA2 and NCEP2) precipitation products were statistically evaluated against the station data for the period 2003–2014. The evaluations were conducted at two spatial (Grid and District) and three temporal (Daily, Monthly and Yearly) scales. In addition to the statistical analysis, the frequency of extreme precipitation events on a daily scale, monsoon and non-monsoon season precipitation variability on a monthly scale, and spatial precipitation variability on a yearly scale were also analyzed. The main results and conclusions are summarized below:

1. ERA5-Land performed the best in Coimbatore, Madurai and Tiruchirappalli, whereas MSWEP performed well at Tuticorin.

2. The accuracy of precipitation products was better at monthly and yearly scales than at the daily scale. Further, the increase of spatial scale from grid to district did not strongly affect the performance of precipitation products.

3. The ability of precipitation products to capture extreme rainfall intensity is different from the statistical metrics. ERA5-Land underestimated the extreme rainfall intensity for all three return periods (2, 5 and 10 years) at the study regions except in Tiruchirappalli. MSWEP captured the closest rainfall intensity in Coimbatore and Madurai, PERSIANN CDR in Tiruchirappalli and ERA5-Land in Tuticorin.

4. Most products underestimated the mean monthly precipitation in monsoon and non-monsoon months. Additionally, the %MAE was higher in non-monsoon months, indicating that these product-based projections for water-scarce periods may be less reliable.





5. The present precipitation algorithms are unable to capture the convective-based low precipitation events in the study regions
(Madurai, Tiruchirappalli, and Tuticorin), leading to underestimation. Hence, algorithm improvisation or prior bias corrections
are needed before considering the evaluated products for future modeling studies.

The evaluation's findings help select the precipitation product for the non-gauged area in agro-hydrological modeling.
Additional accuracy and the selection of a better product for future analysis can be obtained from the event-level assessment
of extreme precipitation events, monsoon, and non-monsoon precipitation. Precipitation is the major source of soil moisture,
which in turn drives food production. Identifying accurate precipitation is one of the underlying factors of the nexus between
food production (agriculture), water, and soil management. Hence, the study has significant implications for managing the
nexus between food production and water and soil management in agricultural regions.

***Data availability.*** In this study, publicly available daily precipitation data are retrieved from multiple climate data sources, as
provided below. Satellite based precipitation product used in this study are the CPC MORPHing technique(CMORPH)
available at National Centres for Environmental Information-Climate Data Records (NCEI-CDR) library (Index of
/data/cmorph-high-resolution-global-precipitation-estimates/access/daily (noaa.gov)), Global Precipitation Measurement
(GPM) and Tropical Rainfall Measuring Mission (TRMM) available at NASA Earth Data search library (Earthdata Search |
Earthdata Search (nasa.gov)), Multi-Source Weighted-Ensemble Precipitation (MSWEP) available at GloH2O library
(MSWEP - GloH2O)  , Precipitation Estimation from Remotely Sensed Information using Artificial Neural Networks –
Climate Data Records (PERSIANN CDR) available at NCEI-CDR library (Index of /data/precipitation-persiann/access
(noaa.gov)), European Centre for Medium Range Weather Forecast (ECMWF) Reanalysis version 5 Land (ERA5-Land)
available at Copernicus Climate Change Service (C3S) Climate Data Store (ERA5-Land hourly data from 1950 to present
(copernicus.eu)), Modern-Era Retrospective analysis for Research and Application version 2 (MERRA 2) available at Global
Modeling                                     and                                     Assimilation                                     Office
library(https://disc.gsfc.nasa.gov/datasets?page=1&subject=Precipitation&source=Models%20MERRA-
2&project=MERRA-2&temporalResolution=1%20day), National Center for Environmental Prediction Reanalysis 2 (NCEP2)
available at NOAA Physical Science Laboratory (NCEP/DOE Reanalysis II: NOAA Physical Sciences Laboratory
NCEP/DOE Reanalysis II).


***Author contributions.*** AS, DK, and NS designed the research. AS performed the data analysis and prepared the manuscript
with substantial contributions from DK, NS and SR.

***Competing interests***. The authors declare that they have no conflict of interest.




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
