# Peer review of "Evaluation of remote sensing and reanalysis based precipitation products for agro-hydrological studies in semi-arid tropics of Tamil Nadu"

_EGUsphere, 2024_

## Author Comment (AC1)

The precipitation products evaluated in the study was chosen based on their underlying theory, performance in similar climatic conditions, available spatial and temporal resolution.

Validity of the Gauge-free evaluation like TC method depend on each dataset being an independent estimate of the truth (Stoffelen 1998). The precipitation products used in the present study have certain overlap in the observation techniques and equipment used to generate the retrieval. For instance, satellite-based products, use measurements from infrared and/or microwave sensors as the basis for their precipitation estimates. Reanalysis and gauge-based products also incorporate measurements from many of the same observation station. This violates the assumption of zero error cross correlation, which can lead to bias in TC estimates (Gruber et al. 2016). Further, considering this assumption would have greatly restricted the products to be considered in the study.

Previous studies evaluating TC method considered products which hasn't been previously evaluated for climatic conditions in India (Duan et al. 2021, Lu et al. 2021). For instance, soil moisture based precipitation product SM2RAIN-ASCAT hasn't been evaluated in similar Indian climatic zones. Further, the product isn't available prior to 2007 for conducting an initial assessment with the station data. Directly evaluating such products can violate the assumptions of TC and lead to higher uncertainty in the estimates.

Another major drawback of this methodology is that it requires log transformation data preprocessing. This requires either to remove the days with zero rainfall (Massari et al. 2017) or to replace with very small value such as $10^{-4}$ (Duan et al. 2021). So far, clear conclusion hasn't been reached for zero precipitation days and this will produce a noticeable impact on RMSE (Lu et al. 2021). As semi arid regions are considered in the present study where there are higher number of days with zero rainfall, this will influence the accuracy of evaluation results. Hence, despite the station data being almost decade old, it will provide the most direct and accurate representation of the study region for evaluation.

**References**

Duan, Z., Duggan, E., Chen, C., Gao, H., Dong, J., & Liu, J. (2021). Comparison of traditional method and triple collocation analysis for evaluation of multiple gridded precipitation products across Germany. Journal of Hydrometeorology, 22(11), 2983-2999.

Gruber, A., and Coauthors, 2016: Recent advances in (soil moisture) triple collocation analysis. *Int. J. Appl. Earth Obs. Geoinf.*, 45, 200–211, https://doi.org/10.1016/j.jag.2015.09.002.

Lu, X., Tang, G., Liu, X., Wang, X., Liu, Y., & Wei, M. (2021). The potential and uncertainty of triple collocation in assessing satellite precipitation products in Central Asia. *Atmospheric Research*, *252*, 105452.

Marshall, G.J., 2002. Trends in Antarctic geopotential height and temperature: A comparison between radiosonde and NCEP–NCAR reanalysis data. J. Clim. 15 (6), 659–674.

Stoffelen, A., 1998: Toward the true near-surface wind speed:Error modeling and calibration using triple collocation. *J. Geophys*. Res., 103, 7755–7766, https://doi.org/10.1029/97JC03180.

---

## Author Comment (AC3)

**General comments:**

1. **Comment:** Station to grid comparison: The authors assert that station to grid comparisons are difficult and conduct 'linear interpolation' from station to grid and from product grid to a common 0.1x0.1 degree comparison. I find the overall description of this vague and am also not entirely convinced that this solves the problem given that a lot of the variability within each grid-cell is due to topography and localized patterns that don't change linearly. I am also concerned/ confused that the authors ony consider stations within the district for the regional/ district comparison. Given the irregular shape additional stations outside the region should also be considered.

**Response:** Previous studies have used interpolated ground station data to evaluate precipitation products (Liu et al., 2015, Duan et al. 2016, Shukla et al. 2019). The study considered stations distributed both inside and on the boundary of the study region, ensuring a comprehensive representation of precipitation within the region. While the irregular shape of the region could suggest including external stations, this was not feasible due to the poor distribution of meteorological stations in surrounding areas during the study period. The surrounding districts lacked research institutes and Agricultural Universities, which were essential for maintaining meteorological stations and providing reliable data. Since the real stations are arranged almost in a regular grid, we can expect low interpolation errors. This is also shown by the LOOCV analysis below.

**Table: LOOCV performance of Linear interpolation of precipitation at monthly timescale**

| Study Region | Mean RMSE (mm) | Percent Mean Absolute Error (%) |
|---|---|---|
| Coimbatore | 39.22 | 36 |
| Madurai | 52.50 | 30 |
| Tiruchirappalli | 39.93 | 29 |
| Tuticorin | 42.32 | 25 |

[Figure]

**Figure: Linearly interpolated grids (0.1°) of the study regions along with the station data.**

In the figure above, red points denote the distribution of station datasets. The grids and the stars represent the linear interpolation. Only grids that are surrounded by at least one rain gauge station were included in the analysis to avoid uncertainties in the analysis. A similar explanation has been addressed to Comment No: 8.

2. **Comment:** Choice of regions: I might have missed that, but why are only some regions compared and not India as a whole?

**Response:** This study investigates the use of climate data products for agricultural analyses, focusing on agriculturally significant semi-arid regions in Tamil Nadu: Coimbatore, Madurai, Tiruchirappalli, and Tuticorin. These regions were chosen because each has a State-owned Agricultural University, which ensures their agricultural representativeness and data availability. Although numerous studies have evaluated precipitation products in India, such evaluations are typically conducted at the state or district levels using weekly or monthly time scales. The accuracy of these assessments often depends on access to ground station data, which is not uniformly available across the country. To overcome this challenge, the present study utilizes high-resolution daily data tailored to the specific agro-hydrologic units under consideration. Including India as a whole would limit the spatial resolution of the evaluation, particularly at the grid level for individual districts. The findings of this study are designed to support field-level experimentation and provide a proof of concept for modelers developing climate data products, with the potential for extrapolation to other regions with similar agro-climatic conditions.

3. **Comment:** Reliance on Tables and many figures to compare: Apart from the Taylor diagrams, the authors have many tables and many figures with subplots that compare values between the products. It is very difficult to keep track of all of these comparisons. It would be good to think about a better way to integrate and present results.

**Response:** In the revised document, we have reduced the number of metrics, which allowed us to consolidate two tables into a single table for better integration. Additionally, the figures in the main document have been streamlined to focus on key timescales: daily (IDF plots), monthly (Percent MAE), and annual (Taylor diagrams, Spatial maps). To improve clarity, the IDF plots now exclude return periods for 2 and 10 years, focusing instead on only 5-year return period.

As a result, the revised manuscript now includes three tables and four figures (with subplots) corresponding to the three timescales, making the comparisons more concise and easier to follow. In this way, we integrated tables and reduced reductant figures for better clarity without loss of information. The sequence of Tables and Figures in the revised manuscript is given at the end of this document.

**Specific comments:**

4. **Comment:** Section 2.2. This section could probably be shortenened to focus on the most important information here.

**Response:** As per the reviewer's suggestion, the information has been reduced to highlight only the most important information.

5. **Comment:** L153: Climate Data Guide, 2024 is not an appropriate citation for the datasets since the CDG is not the primary source of the data but a guide for data users by UCAR.

**Response:** As per reviewer's suggestion, the Climate Data Guide referenced has now been revised to Huffman et al. (2023), as mentioned on the webpage.

6. **Comment:** Figure 1: The regular gridded station distribution seems to be an error in the figure?

**Response:** The appearance of a regular gridded station distribution in Figure 1 is not an error, but rather a result of how the data were selected for homogenous distribution within the districts. The datasets used in this study were collected from the Public Works Department (PWD) of Tamil Nadu, which collects and maintains meteorological data for the entire state. Additionally, each selected study region has a State-owned Agricultural University with many crop-specific research institutes, including the Wheat Research Station in Coimbatore, located at a higher altitude. These institutes provide both agro-meteorological and hydro-meteorological data, enabling coverage even in mountainous regions. According to Rajeev et al. (2005), the southern Peninsular region, where the study is located, has a higher density of meteorological stations. When these datasets are combined at the state level, the station data may appear to be distributed in a regular grid, even though the actual distribution is more varied.

7. **Comment:** L240: " For quality reasons, the years 2005 and 2010 were excluded from the present study" > Please explain

**Response:** There were large data gaps in those two years, which means more missing than available data. Instead of filling the gaps with uncertain results using imputation methods, we decided to skip the years.

8. **Comment:** L253: "The interpolated 0.1degree station dataset was used as ground truth to evaluate all the other precipitation products" > See my general comment. Also, it would be really good if the authors could come up with a way to provide any kind of quality measure for this. For example, the authors could have reserved some stations for verification of that methodolody. Or conduct a leave-one-out cross-validation to assess how well the interpolated data reflects actual precipitation in that location.

**Response:** As per the Reviewer's suggestion, Leave-One-Out Cross Validation was conducted to assess how well the interpolated data reflected the actual precipitation. LOOCV was performed on linear interpolation based on assuming that the value of a station was unknown. The unknown station was estimated from the value of the neighboring stations based on linear interpolation. The analysis was systematically carried out in Python. Since the real stations are arranged almost in a regular grid, we can expect low interpolation errors. This is also shown by the LOOCV analysis below.

**Table: LOOCV performance of Linear interpolation of precipitation at monthly timescale**

| Study Region | Mean RMSE (mm) | Percent Mean Absolute Error (%) |
|---|---|---|
| Coimbatore | 39.22 | 36 |
| Madurai | 52.50 | 30 |
| Tiruchirappalli | 39.93 | 29 |
| Tuticorin | 42.32 | 25 |

9. **Comment:** L320: It would be good to also provide MAE as a percentage value of mean precipitation.

**Response:** This will be added in the supplementary table of the revised manuscript.

10. **Comment:** Figure 2: I am a bit confused with this figure because alls of these lines seem to be perfectly straigt on a log-log plot and that is something that I would not have expected. It is also not possible to always see all lines.

**Response:** As the IDF graph was perfectly straight on the log-log plot, we now revised it to a scatter plot (without taking log on both the x and y axes). We opted for this plotting compared to the log-log plot because similar studies followed this technique (Ombadi et al. 2018, Ghebreyesus & Sharif, 2021). Also, the time scale is now of more significance concerning hydrometeorological events. Since some of the products had values close to each other, variations in terms of line style and color are made in the revised graph to increase the visibility of lines which is given at the end of the document (Figure 2). Also, to highlight the performance for a single return period and avoid redundancy, sub-plots of Return periods 2 and 10 have been removed.

11. **Comment:** Figures 3-6: I was initailly confused by these figures. I guess the key message here would be, how the errors compare between monosson and non-monsoon season, but for that, the reader has to do their own math.

**Response:** The stacked plots of monsoon and non-monsoon precipitation (Figures 3-6) in the original manuscript have now been revised to column plots for better interpretation. The revised plots now convey the variation of precipitation in both the monthly means along with its %MAE (Figure. 3, given at the end of the document).

12. **Comment:** Figure 8-11: These should contain the station locations to better understand the interpolation.

**Response:** The station data in Figures 8-11 have now been revised, and they contain station data, representing the intra-region precipitation variation and interpolation.

13. **Comment:** L490: "ERA5-Land produced the closest approximation to the station data (Fig. 8). " > It would be good to back this up with some quantittative quality measure rather than a qualitative comparison.

**Response:** As per the reviewer's suggestion, the following quantitative comparison will be added: Based on the correlation values and RMSE given in the Table. 5, along with the Figure 8, it is concluded that ERA5Land produces a close estimate of the station data. Similar explanations will also be added for the other comparisons also.

14. **Comment:** Section 5: This should start with a discussion of results including general patterns and limitations of the study. Then followed by

**Response:** The above-mentioned section is deleted from the revised manuscript, and the discussion section starts with a discussion of the study.

**Revised Tables and Figures:**

**Table 3.** Daily precipitation characteristics of Coimbatore, Madurai, Tiruchirappalli and Tuticorin

[revised manuscript text omitted]

**Madurai**

[Figure]

**Figure 9.** Mean annual precipitation for the Station data at Madurai. *Red dots represent the station data and stars denote the linearly interpolated grids. As the inclusion of station data is relevant information for station data spatial maps, it will be revised as given above and included with the other spatial maps in the revised manuscript.*

[Figure]

**Figure 10.** Mean annual precipitation for the Station data at Tiruchirapalli. *Red dots represent the station data and stars denote the linearly interpolated grids. As the inclusion of station data is relevant information for station data spatial maps, it will be revised as given above and included with the other spatial maps in the revised manuscript.*

[Figure]

**Figure 11.** Mean annual precipitation for the Station data at Tuticorin. *Red dots represent the station data and stars denote the linearly interpolated grids. As the inclusion of station data is relevant information for station data spatial maps, it will be revised as given above and included with the other spatial maps in the revised manuscript.*

---

## Author Comment (AC4)

1. **Comment**: **L 38:** Citation required

**Response:** The following reference will be added to the revised manuscript: Government of Tamil Nadu, (2022a).

2. **Comment**: **L 46:** Citation required

**Response:** The following references will be added to the revised manuscript: Arjune and Kumar (2023), Balaganesh *et al*. (2020), Gardas *et al*. (2018), Malaiarasan *et al*. (2021).

3. **Comment**: **L 53**: Citation required

**Response:** The following references will be added to the revised manuscript: Radhakrishnan et al. (2024), Lalmuanzuala et al. (2023), Paramasivam (2023)

4. **Comment**: Citation required

**Response:** The reference was taken from IPCC Sixth Assessment Report (IPCC, 2023)

5. **Comment**: **L 77**: Table

**Response:** Required changes will be made in the revised manuscript.

6. **Comment:** This part is not required in the introduction, and it can be included in the dataset

**Response:** As suggested by the reviewer, the section will be removed from the Introduction and included in the 'Dataset'.

7. **Comment:**

**Response:** Required changes will be made in the revised manuscript.

8. **Comment:** L **119**: Study region

**Response:** To avoid redundancy in the information presented, Section 2.1 and Section 3.1 will be merged in the revised manuscript.

9. **Comment**: **L 127:** long-term variability and change can also be represented to show the changes in pattern

**Response:** The precipitation pattern showed inter-annual variability, as shown in the figure below. The year 2008 recorded the highest precipitation, whereas the year 2012 indicated the lowest. The results of the Mann-Kendall test (MK-test) indicated no significant trend in the annual precipitation for these four regions.

[Figure]

10. **Comment: L 144:** Highest spatial resolution

**Response:** We selected products with spatial resolutions between 0.1° and 0.25°. While reanalysis products typically have coarser resolutions, we included MERRA2 and NCEP2 in our selection due to their unique advantages. Both products provide global coverage on a daily timescale and incorporate significant advancements. For example, NCEP2 improves precipitation parameterizations using cloud-top cooling (Kanamitsu et al., 2002), while MERRA2 integrates atmospheric aerosols into its analysis (Bosilovich et al., 2015).

11. **Comment**: **L 235:** Methodology can be as section 2, and the study region should come as sub-section. Then, tehre wont be any repetion of the sections. Sectio 2.1 and 3.1 are repeating, these can be combined together

**Response:** We will merge sections 2.1 and 3.1 to avoid redundancy. So, the revised section now looks like,

1. Introduction
2. Methodology
   2.1 Study region and Ground station
   2.2 Datasets
   2.3 Comparison of ground data with satellite and observational reanalysis-based data
       2.3.1   Grid scale comparison
       2.3.2   District scale comparison
   2.4 Evaluation metrics
3. Results
4. Discussion
5. Conclusion

**12. Comment: L 251:** what was the duration of data used to compare

**Response:** The study used 10 years of precipitation data for evaluation of the products. Many previous studies have used the approach of interpolating station data onto a grid and then conducting evaluations using the grid-to-grid method (Duan et al. 2016, Liu et al. 2015, Shukla et al. 2019). The present study adopted this approach to align with the commonly used practice.

**13. Comment: L 261:** Evaluation can be included as a separate sub-heading as section 3.3

**Response:** 'Evaluation metrics' will be included as a separate sub-heading in the revised manuscript. Section in the revised manuscript as response to Comment: 12.

**14. Comment: L 734:** The bullet points can be avoided in conclusion

**Response:** We thank the reviewer for the suggestion. Conclusion will be presented as a paragraph in the revised manuscript.

**15. Comment: L 747:** So the study can not generalise the use if any precipiattion data. Is ERA5 can be adopted to other similar agro-climatic conditions? What is the

controbution of this study towards other data-sparse regions in India, other than the selected 4 locations?good to include future prospect.

**Response:** The following section on future prospects will be included in the revised manuscript:

The results of the study can be useful for the districts falling within the same agro-climatic regions. The state of the Tamil Nadu is divided into 7 agroclimatic zones. The present study includes 3 important agricultural zones. The results can be used for other districts within the same agroclimatic zones. The findings of this study are designed to support field-level experimentation and also provide a proof of concept for modelers developing climate data products, with the potential for extrapolation to other regions with similar agro-climatic conditions.

**References:**

Arjune, S., & Kumar, V. S. (2023, February). Precision Agriculture: Influencing factors and challenges faced by farmers in delta districts of Tamil Nadu. In *2022 OPJU International Technology Conference on Emerging Technologies for Sustainable Development (OTCON)* (pp. 1-6). IEEE.

Balaganesh, G., Malhotra, R., Sendhil, R., Sirohi, S., Maiti, S., Ponnusamy, K., & Sharma, A. K. (2020). Development of composite vulnerability index and district level mapping of climate change induced drought in Tamil Nadu, India. *Ecological Indicators*, *113*, 106197.

Bosilovich, M. G., Lucchesi, R., and Suarez, M.: MERRA-2: File specification, 2015.

Duan, Z., Liu, J., Tuo, Y., Chiogna, G., and Disse, M.: Evaluation of eight high spatial resolution gridded precipitation products in Adige Basin (Italy) at multiple temporal and spatial scales, Science of the Total Environment, 573, 1536–1553, https://doi.org/10.1016/j.scitotenv.2016.08.213, 2016.

Gardas, B. B., Raut, R. D., & Narkhede, B. (2018). Evaluating critical causal factors for post-harvest losses (PHL) in the fruit and vegetables supply chain in India using the DEMATEL approach. *Journal of cleaner production*, *199*, 47-61.

Government of Tamil Nadu: Statistical Handbook 2020-21: Sub chapter: Rainfall, 2022a.

IPCC, 2023: Summary for Policymakers. In: Climate Change 2023: Synthesis Report. Contribution of Working Groups I, II and III to the Sixth Assessment Report of the Intergovernmental Panel on Climate Change [Core Writing Team, H. Lee and J. Romero (eds.)]. IPCC, Geneva, Switzerland, pp. 1-34, doi: 10.59327/IPCC/AR6-9789291691647.001

Kanamitsu, M., Ebisuzaki, W., Woollen, J., Yang, S.-K., Hnilo, J. J., Fiorino, M., and Potter, G. L.: Ncep–doe amip-ii reanalysis (r-2), Bull Am Meteorol Soc, 83, 1631–1644, 2002.

LALMUANZUALA, B., SATHYAMOORTHY, N., KOKILAVANI, S., JAGADEESWARAN, R., & KANNAN, B. (2023). Drought analysis in southern region of Tamil Nadu using meteorological and remote sensing indices. *MAUSAM, 74*(4), 973-988.

Liu, J., Duan, Z., Jiang, J., Zhu, A.X., 2015. Evaluation of three satellite precipitation products TRMM 3B42, CMORPH, and PERSIANN over a subtropical watershed in China. Adv. Meteorol. 2015 https://doi.org/10.1155/2015/151239.

Malaiarasan, U., Paramasivam, R., & Felix, K. T. (2021). Crop diversification: determinants and effects under paddy-dominated cropping system. *Paddy and Water Environment, 19*, 417-432.

Paramasivam P (2023) Hundreds stranded as parts of India's Tamil Nadu flooded after heavy rain, *Reuters*: Hundreds stranded as parts of India's Tamil Nadu flooded after heavy rain | Reuters

Radhakrishnan, S., Duraisamy Rajasekaran, S. K., Sujatha, E. R., & Neelakantan, T. R. (2024). A Comparative Study on 2015 and 2023 Chennai Flooding: A Multifactorial Perspective. *Water, 16*(17), 2477.

Shukla, A.K., Ojha, C.S.P., Singh, R.P., Pal, L., Fu, D., 2019. Evaluation of TRMM precipitation dataset over Himalayan Catchment: the upper Ganga Basin, India. Water (Switzerland) 11 (3). https://doi.org/10.3390/w11030613.

---

## Author Response (AR3)

We express our gratitude to the referees and the handling editor for their valuable time in helping improve this manuscript. In our opinion, the referees' input will considerably improve the revised manuscript. Here will outline all changes we will add to the initial manuscript. This is our initial response. Planned activities are also included in the document and has all been implemented.

- Restructuring subsections "Study region and Ground station"
- The 'Datasets' section will be revised only highlighting important information
- Inclusion of methodology – Grid and District scale evaluation, Monsoon and Non-monsoon month evaluation
- Inclusion of all the equation numbers and abbreviations in the methodology
- Consistent use of Acronyms, e.g. 'GPM-IMERG'
- Tables are revised to include only Correlation Coefficient (CC), Root Mean Square Error (RMSE) and RB (Relative Bias)
- Revised Figure. 2 to include only IDF for 5-year Return Period
- Restructuring the 'Discussion' section by describing the results with related studies followed by the limitations of the present study.
- Conclusion will be revised and include an outlook of the present study's impact in other data-scarce regions in India.

We added a response to the reviewer's individual comments in three separate sections below.

**Responses to Referee 1's Comments**

**Specific comments :**

1. **Comment:** Line 87. When you mention GPM, are these IMERG data? GPM-IMERG is generally used.

**Response:** Yes, it is IMERG final run products. As per reviewer's suggestion, GPM has been revised to GPM-IMERG in the entire revised manuscript.

2. **Comment:** Line 140. Are the rain gauges (69 rain gauges) used for validation included in the GPCC? If not, please specify.

**Response:** The documentation for confirming whether the rain gauge data are included in GPCC is not currently accessible to everyone. The present study used data from the State Government of Tamil Nadu's Public Works Department (PWD). PWD is responsible for executing development works in the state and maintains a repository of meteorological stations.

GPCC, on the other hand, collects data from meteorological stations established by the central government of a country, which is the India Meteorological Department (IMD) for India. As IMD and the Tamil Government's PWD operate at different levels, there is no evidence of transferring these station data at the state level to the central government to include it in GPCC. It is also not mentioned in the GPCC contributors list - Wetter und Klima - Deutscher Wetterdienst - Our services - New data contributions to GPCC

3. **Comment:** In Table 2, IMERG-GPM can go up to hourly.

**Response:** Yes, IMERG-GPM also has sub-daily and hourly data. The present study uses only daily resolution, which is mentioned as 'daily' in the Table. To bring more clarity, '#Temp' will be revised to 'Temporal resolution used in the present study' in the revised manuscript (Line 258).

4. **Comment:** Why is there redundancy on the study regions in the methodology and study area in part 2?

**Response:** We thank the reviewer for observing the redundancy in the structure of the manuscript. We will merge sections 2.1 and 3.1. Considering Reviewer 3's comment number 11, Methodology has been added as Section 2. So, the revised section now looks like,

1. Introduction
2. Methodology
   2.1 Study regions
   2.2 Datasets
       2.2.1 Ground stations
       2.2.2 Satellite and Reanalysis-based Precipitation Products
   2.3 Comparison of ground data with satellite and observational reanalysis-based data
       2.3.1 Evaluation at grid scale

**5.  Comment:** Figure 1. It's a bit strange that the ground stations are very aligned. How is this possible? There are rain gauges in the mountains even though the area is supposed to be difficult to access.

**Response:** The appearance of a regular gridded station distribution is a result of how the data were selected for homogenous distribution within the districts. The datasets used in this study were collected from the Public Works Department (PWD) of Tamil Nadu, which collects and maintains meteorological data for the entire state. Additionally, each selected study region has a State-owned Agricultural University with many crop-specific research institutes, including the Wheat Research Station in Coimbatore, located at a higher altitude. These institutes provide both agro-meteorological and hydro-meteorological data, enabling coverage even in mountainous regions. According to Rajeev et al. (2005), the southern Peninsular region, where the study is located, has a higher density of meteorological stations. When these datasets are combined at the state level, the station data may appear to be distributed in a regular grid, even though the actual distribution is more varied.

Suitable explanations are added to the revised manuscript (Line180-Line 191).

A similar response has been addressed to Reviewer 2's Comment No: 6.

**6.  Comment:** Line 253. Doesn't this already introduce a large bias for the data used for validation? In other studies, "point-gridded" is used. Because the location of rain gauges most often does not coincide with gridded precipitation products (GPP) grid centroids, a second strategy was implemented: the point-gridded approach. In practice, a cell is delineated around each rain gauge (cell size of 0.04, 0.05, or 0.1°… depending on the GPP; Table. 2). Then, the rainfall value in those new cells was estimated as the area-weighted mean (max. 4) of the GPP grid cells overlapping with the new cell.

**Response:** Previous studies have used interpolated ground station data to evaluate precipitation products (Liu et al., 2015, Duan et al. 2016, Shukla et al. 2019). Since the real stations are arranged almost in a regular grid, we can expect low interpolation errors. This is also shown by the LOOCV analysis below.

We performed a LOOCV analysis on the linear interpolated data to check for bias or uncertainty in the data. LOOCV was performed on linear interpolation based on assuming that the value of a station was unknown. The unknown station was estimated from the value of the neighbouring stations based on linear interpolation. The analysis was systematically carried out in Python. The results of the LOOCV analysis represent the mean RMSE at a monthly timescale, which includes both monsoon and non-monsoon months. The RMSE and %MAE of LOOCV analysis produced lower values, indicating minimal uncertainty due to the linear interpolation. Hence, we proceeded with linear interpolation of station data before grid-to-grid comparison.

**Table: LOOCV performance of Linear interpolation of precipitation at monthly timescale**

| Study Region | Mean RMSE (mm) | Percent Mean Absolute Error (%) |
|---|---|---|
| Coimbatore | 39.22 | 36 |
| Madurai | 52.50 | 30 |
| Tiruchirappalli | 39.93 | 29 |
| Tuticorin | 42.32 | 25 |

[Figure]

**Figure: Linearly interpolated grids (0.1°) of the study regions along with the station data.** In the figure above, red points denote the distribution of station datasets. The grids and the stars represent the linear interpolation. Only grids that are surrounded by at least one rain gauge station were included in the analysis to avoid uncertainties in the analysis.

The LOOCV explanation has been included in the supplementary section of the revised manuscript.

**Results.**

7. **Comment:** Grid scale/district scale method are not really mentioned in the methodology part.

**Response:** We agree with the reviewer's comment that the methodology does not clearly mention the grid scale/district scale. The following section has been added to the revised manuscript (Line 273-Line 286):

2.3.1 Evaluation at grid scale

The interpolated station dataset was used as ground truth to evaluate all the other downscaled precipitation products at 0.1⬚ spatial scales. The evaluation was considered for grids having atleast one rain gauge in the corresponding grid or in the surrounding grids.

2.3.2 Evaluation at district scale

The gridded precipitation from the grids considered at the grid-scale evaluation were averaged to estimate the district precipitation amount. This way, district-level precipitation was estimated for the

station data and the precipitation products. The developed district-level data was then evaluated, keeping the station value as a ground truth for the precipitation products.

**8. Comment:** Monsoon/Non-Monsoon not clear in the methodology.

**Response:** We agree with the reviewer's comment that Monsoon/Non monsoon description is not sufficiently mentioned in the methodology. The following section is added to the revised manuscript (Line 282-Line 285).

Since the study region has abundant rainfall during the monsoon months, which is significantly higher than the rest of the months, the study evaluated the performance of the products during the monsoon and non-monsoon seasons. The study region falls under the North East monsoon zone from October to December (Table 1). So, average precipitation values of October, November, and December were taken as monsoon months. The average precipitation for the rest of the year, i.e from January to September, was taken as Non-monsoon months.

**Discussion**

**9. Comment:** The first paragraph seems out of place and confuses the reader. Start directly by discussing the results.

**Response:** We removed the first paragraph in the revised manuscript and will start by discussing the results.

**10. Comment:** Line 665. Why are ERA-5 Land and MSWEP performing better than others? Algorithms used? Reanalyzed products? Is this the case for other studies?

**Response:** ERA5-Land performs better than other products as it was developed based on replay of land component with H-TESSEL model. This brings a special improvement compared to ERA5 and is producing best results compared to other products used in this study (Muñoz-Sabater et al., 2021). Similar results have been reported by Kolluru et al. (2020). MSWEP uses ensemble of different precipitation products which contributed to its best estimations; Nair and Indu (2017) also reported similar results for MSWEP performance in India.

As per the reviewer's suggestions, required amendments were made in Line 656 – Line 660 and Line 673 -Line 675 of the revised manuscript.

**11. Comment:** Line 687. From what rainfall intensity does ERA-5 Land struggle to detect?

**Response:** ERA5-Land struggles to detect precipitation intensity of more than 1 mm /hour in Coimbatore and more than 5 mm/hour in Madurai, as observed in Fig. 2. This underestimation might probably be due to the inability of the product to capture lightning-associated heavy rainfall, which was reported in ERA5 by Kumar et al. (2024).

The above-mentioned explanation has been added to Line 681-Line 684 of the revised manuscript.

**12. Comment:** Line 694 – 704. The entire paragraph should be dispatched into other paragraphs to properly explain the reasons for the performance of precipitation products compared to others."

**Response:** In accordance with the reviewer's suggestion, the required explanations are made in the revised manuscript, which explains the previous studies and the product's specific algorithm supporting (or) contributing to the present study's result. Revisions were made in Line 656-Line 660, Line 662-Line 665, Line 673-Line 675, Line 682-Line 684.

**Technical comments :**

**13. Comment:** Line 266. What do all the terms mean? *Pi, …*

**Response:** Pi denotes the value of the precipitation Product at time I, whereas $\bar{P}$ denotes the average. The explanations are added in the revised manuscript in Line 293-Line 295.

**14. Comment:** Line 283. Equation 7 not mentioned in the text.

**Response:** Equation 7 is added in the revised manuscript in Line 310.

**15. Comment:** Same for Eq. 8, 9, 10

**Response:** Equations 8,9 and 10 are added in Line 313, 314 and 315, respectively

**16. Comment:** Line 290. Equation 11 instead of Equation 7.

**Response:** We thank the reviewer for the kind observation. In revised manuscript, Equation 11 is now mentioned in Line 320.

**17. Comment:** Equation 12 not mentioned in the text.

**Response:** Equation 12 is added in Line 328 of the revised manuscript.

**18. Comment:** Line 302. Equation 13 instead of Equation 9.

**Response: :** We thank the reviewer for the kind observation. Equation 13 is mentioned in Line 332.

**Comment:** Review all Equation numbers.

**Response:** Reviewed and revised in the revised manuscript

**Additional References**

Duan, Z., Liu, J., Tuo, Y., Chiogna, G., and Disse, M.: Evaluation of eight high spatial resolution gridded precipitation products in Adige Basin (Italy) at multiple temporal and spatial scales, Science of the Total Environment, 573, 1536‑1553, https://doi.org/10.1016/j.scitotenv.2016.08.213, 2016.

Liu, J., Duan, Z., Jiang, J., Zhu, A.X., 2015. Evaluation of three satellite precipitation products TRMM 3B42, CMORPH, and PERSIANN over a subtropical watershed in China. Adv. Meteorol. 2015 https://doi.org/10.1155/2015/151239.

Muñoz-Sabater, J., Dutra, E., Agustí-Panareda, A., Albergel, C., Arduini, G., Balsamo, G., Boussetta, S., Choulga, M., Harrigan, S., and Hersbach, H.: ERA5-Land: A state-of-the-art global reanalysis dataset for land applications, Earth Syst Sci Data, 13, 4349–4383, 2021.

Nair, A. S. and Indu, J.: Performance assessment of multi-source weighted-ensemble precipitation (MSWEP) product over India, Climate, 5, 2, 2017.

Shukla, A.K., Ojha, C.S.P., Singh, R.P., Pal, L., Fu, D., 2019. Evaluation of TRMM precipitation dataset over Himalayan Catchment: the upper Ganga Basin, India. Water (Switzerland) 11 (3). https://doi.org/10.3390/w11030613.

Reddy, N. M. and Saravanan, S.: Evaluation of the accuracy of seven gridded satellite precipitation products over the Godavari River basin, India, International Journal of Environmental Science and Technology, 20, 10179‑10204, https://doi.org/10.1007/s13762-022-04524-x, 2023.

Kolluru, V., Kolluru, S., & Konkathi, P. (2020). Evaluation and integration of reanalysis rainfall products under contrasting climatic conditions in India. *Atmospheric Research*, *246*, 105121.

**Responses to Referee 2's Comments**

**General comments:**

1. **Comment:** Station to grid comparison: The authors assert that station to grid comparisons are difficult and conduct 'linear interpolation' from station to grid and from product grid to a common 0.1x0.1 degree comparison. I find the overall description of this vague and am also not entirely convinced that this solves the problem given that a lot of the variability within each grid-cell is due to topography and localized patterns that don't change linearly. I am also concerned/ confused that the authors ony consider stations within the district for the regional/ district comparison. Given the irregular shape additional stations outside the region should also be considered.

**Response:** Previous studies have used interpolated ground station data to evaluate precipitation products (Liu et al., 2015, Duan et al. 2016, Shukla et al. 2019). The study considered stations distributed both inside and on the boundary of the study region, ensuring a comprehensive representation of precipitation within the region. While the irregular shape of the region could suggest including external stations, this was not feasible due to the poor distribution of meteorological stations in surrounding areas during the study period. The surrounding districts lacked research institutes and Agricultural Universities, which were essential for maintaining meteorological stations and providing reliable data. Since the real stations are arranged almost in a regular grid, we can expect low interpolation errors. This is also shown by the LOOCV analysis below.

**Table: LOOCV performance of Linear interpolation of precipitation at monthly timescale**

| Study Region | Mean RMSE (mm) | Percent Mean Absolute Error (%) |
|---|---|---|
| Coimbatore | 39.22 | 36 |
| Madurai | 52.50 | 30 |
| Tiruchirappalli | 39.93 | 29 |
| Tuticorin | 42.32 | 25 |

[Figure]

[Figure]

[Figure]

**Figure: Linearly interpolated grids (0.1°) of the study regions along with the station data.**

In the figure above, red points denote the distribution of station datasets. The grids and the stars represent the linear interpolation. Only grids that are surrounded by at least one rain gauge station were included in the analysis to avoid uncertainties in the analysis. A similar explanation has been addressed to Comment No: 8.

2. **Comment:** Choice of regions: I might have missed that, but why are only some regions compared and not India as a whole?

**Response:** This study investigates the use of climate data products for agricultural analyses, focusing on agriculturally significant semi-arid regions in Tamil Nadu: Coimbatore, Madurai, Tiruchirappalli, and Tuticorin. These regions were chosen because each has a State-owned Agricultural University, which ensures their agricultural representativeness and data availability. Although numerous studies have evaluated precipitation products in India, such evaluations are typically conducted at the state or district levels using weekly or monthly time scales. The accuracy of these assessments often depends on access to ground station data, which is not uniformly available across the country. To overcome this challenge, the present study utilizes high-resolution daily data tailored to the specific agro-hydrologic units under consideration. Including India as a whole would limit the spatial resolution of the evaluation, particularly at the grid level for individual districts. The findings of this study are designed to support field-level experimentation and provide a proof of concept for modelers developing climate data products, with the potential for extrapolation to other regions with similar agro-climatic conditions.

3. **Comment:** Reliance on Tables and many figures to compare: Apart from the Taylor diagrams, the authors have many tables and many figures with subplots that compare values between the products. It is very difficult to keep track of all of these comparisons. It would be good to think about a better way to integrate and present results.

**Response:** In the revised document, we have reduced the number of metrics, which allowed us to consolidate two tables into a single table for better integration. Additionally, the figures in the main document have been streamlined to focus on key timescales: daily (IDF plots), monthly (Percent MAE), and annual (Taylor diagrams, Spatial maps). To improve clarity, the IDF plots now exclude return periods for 2 and 10 years, focusing instead on only 5-year return period.

As a result, the revised manuscript now includes three tables and eight figures (with subplots) corresponding to the three timescales, making the comparisons more concise and easier to follow. In this way, we integrated tables and reduced reductant figures for better clarity without loss of information.

**Specific comments:**

4. **Comment:** Section 2.2. This section could probably be shortenened to focus on the most important information here.

**Response:** As per the reviewer's suggestion, the information has been reduced to highlight only the most important information (Line 192 – Line 246).

5. **Comment:** L153: Climate Data Guide, 2024 is not an appropriate citation for the datasets since the CDG is not the primary source of the data but a guide for data users by UCAR.

**Response:** As per reviewer's suggestion, the Climate Data Guide referenced has now been revised to Huffman et al. (2023), as mentioned on the webpage (Line 209).

6. **Comment:** Figure 1: The regular gridded station distribution seems to be an error in the figure?

**Response:** The appearance of a regular gridded station distribution in Figure 1 is not an error, but rather a result of how the data were selected for homogenous distribution within the districts. The datasets used in this study were collected from the Public Works Department (PWD) of Tamil Nadu, which collects and maintains meteorological data for the entire state. Additionally, each selected study region has a State-owned Agricultural University with many crop-specific research institutes, including the Wheat Research Station in Coimbatore, located at a higher altitude. These institutes provide both agro-meteorological and hydro-meteorological data, enabling coverage even in mountainous regions. According to Rajeev et al. (2005), the southern Peninsular region, where the study is located, has a higher density of meteorological stations. When these datasets are combined at the state level, the

station data may appear to be distributed in a regular grid, even though the actual distribution is more varied.

7. **Comment:** L240: " For quality reasons, the years 2005 and 2010 were excluded from the present study" > Please explain

**Response:** There were large data gaps in those two years, which means more missing than available data. Instead of filling the gaps with uncertain results using imputation methods, we decided to skip the years.

Additional explanations are added in Lines 189-191 of the revised manuscript.

8. **Comment:** L253: "The interpolated 0.1degree station dataset was used as ground truth to evaluate all the other precipitation products" > See my general comment. Also, it would be really good if the authors could come up with a way to provide any kind of quality measure for this. For example, the authors could have reserved some stations for verification of that methodolody. Or conduct a leave-one-out cross-validation to assess how well the interpolated data reflects actual precipitation in that location.

**Response:** As per the Reviewer's suggestion, Leave-One-Out Cross Validation was conducted to assess how well the interpolated data reflected the actual precipitation. LOOCV was performed on linear interpolation based on assuming that the value of a station was unknown. The unknown station was estimated from the value of the neighbouring stations based on linear interpolation. The analysis was systematically carried out in Python. The results of the LOOCV analysis represent the mean RMSE at a monthly timescale, which includes both monsoon and non-monsoon months. The RMSE and %MAE of LOOCV analysis produced lower values, indicating minimal uncertainty due to the linear interpolation. Hence, we proceeded with linear interpolation of station data before grid-to-grid comparison.

**Table: LOOCV performance of Linear interpolation of precipitation at monthly timescale**

| Study Region | Mean RMSE (mm) | Percent Mean Absolute Error (%) |
|---|---|---|
| Coimbatore | 39.22 | 36 |
| Madurai | 52.50 | 30 |
| Tiruchirappalli | 39.93 | 29 |
| Tuticorin | 42.32 | 25 |

9. **Comment:** L320: It would be good to also provide MAE as a percentage value of mean precipitation.

**Response:** This has been added in the supplementary table of the revised manuscript (Table S2, S3, S4).

10. **Comment:** Figure 2: I am a bit confused with this figure because alls of these lines seem to be perfectly straigt on a log-log plot and that is something that I would not have expected. It is also not possible to always see all lines.

**Response:** As the IDF graph was perfectly straight on the log-log plot, we now revised it to a scatter plot (without taking log on both the x and y axes). We opted for this plotting compared to the log-log plot because similar studies followed this technique (Ombadi et al. 2018, Ghebreyesus & Sharif, 2021). Also, the time scale is now of more significance concerning hydrometeorological events. Since some of the products had values close to each other, variations in terms of line style and color are made in the revised graph to increase the visibility of lines. Also, to highlight the performance for a single return period and avoid redundancy, sub-plots of Return periods 2 and 10 have been removed.

Figure 2 in the revised manuscript highlights all the changes explained above.

11. **Comment:** Figures 3-6: I was initailly confused by these figures. I guess the key message here would be, how the errors compare between monosson and non-monsoon season, but for that, the reader has to do their own math.

**Response:** The stacked plots of monsoon and non-monsoon precipitation (Figures 3-6) in the original manuscript have now been revised to column plots for better interpretation. The revised plots now convey the variation of precipitation in both the monthly means along with its %MAE.

Figure. 3 in the revised manuscript highlights all the revisions explained above.

12. **Comment:** Figure 8-11: These should contain the station locations to better understand the interpolation.

**Response:** The station data in Figures 5-8 of the revised manuscript contain ground stations (represented by red dots) and interpolated grids (represented by black stars), for indicating the intra-region precipitation variation and interpolation.

13. **Comment:** L490: "ERA5-Land produced the closest approximation to the station data (Fig. 8). " > It would be good to back this up with some quantittative quality measure rather than a qualitative comparison.

**Response:** As per the reviewer's suggestion, quantitative justification for the spatial plot comparison is added in the revised manuscript (Line 534, Line 542 and Line 548).

14. **Comment:** Section 5: This should start with a discussion of results including general patterns and limitations of the study. Then followed by

**Response:** The above-mentioned section is deleted from the revised manuscript, and the discussion section starts with a discussion of the study.

**Responses to Referee 3's Comments**

1. **Comment**: **L 38:** Citation required

**Response:** The following reference will be added to the revised manuscript: Government of Tamil Nadu, (2022a) in Line 40.

2. **Comment**: **L 46:** Citation required

**Response:** The following references will be added to the revised manuscript: Arjune and Kumar (2023), Balaganesh *et al.* (2020), Gardas *et al.* (2018), Malaiarasan *et al.* (2021) in Line 47.

3. **Comment**: **L 53**: Citation required

**Response:** The following references will be added to the revised manuscript: Radhakrishnan et al. (2024), Lalmuanzuala et al. (2023), Paramasivam (2023) in Line 58.

4. **Comment**: Citation required

**Response:** The reference was taken from IPCC Sixth Assessment Report (IPCC, 2023) and is now mentioned in Line 58 of the revised manuscript.

5. **Comment**: **L 77**: Table

**Response:** Required changes are made in the revised manuscript (Section 2.2.2, Line 193-Line 246).

6. **Comment:** This part is not required in the introduction, and it can be included in the dataset

**Response:** This part was added to the introduction to cover the different products available for the study location. Of this, only selected products were chosen for evaluation, which is mentioned in the 'Datasets' section.

7. **Comment:**

**Response:** Required changes will be made in the revised manuscript (Line 116).

8. **Comment:** L **119**: Study region

**Response:** To avoid redundancy in the information presented, Section 2.1 and Section 3.1 are merged in the revised manuscript.

9. **Comment**: **L 127:** long-term variability and change can also be represented to show the changes in pattern

**Response:** The precipitation pattern showed inter-annual variability, as shown in the figure below. The year 2008 recorded the highest precipitation, whereas the year 2012 indicated the lowest. The results of the Mann-Kendall test (MK-test) indicated no significant trend in the annual precipitation for these four regions.

[Figure]

Additional explanations are added in Line 127-128 and the MK values are presented in Table 1.

10. **Comment: L 144:** Highest spatial resolution

**Response:** We selected products with spatial resolutions between 0.1° and 0.25°. While reanalysis products typically have coarser resolutions, we included MERRA2 and NCEP2 in our selection due to their unique advantages. Both products provide global coverage on a daily timescale and incorporate significant advancements. For example, NCEP2 improves precipitation parameterizations using cloud-top cooling (Kanamitsu et al., 2002), while MERRA2 integrates atmospheric aerosols into its analysis (Bosilovich et al., 2015).

The above explanation is now added in the revised manuscript in Line 194 -L 197.

11. **Comment**: **L 235:** Methodology can be as section 2, and the study region should come as sub-section. Then, tehre wont be any repetion of the sections. Sectio 2.1 and 3.1 are repeating, these can be combined together

**Response:** We will merge sections 2.1 and 3.1 to avoid redundancy. So, the revised section now looks like,

6. Introduction
7. Methodology
   7.1 Study regions
   7.2 Datasets
       7.2.1    Ground stations
       7.2.2    Satellite and Reanalysis-based Precipitation Products
   7.3 Comparison of ground data with satellite and observational reanalysis-based data

**12. Comment: L 251:** what was the duration of data used to compare

**Response:** The study used 10 years of precipitation data for evaluation of the products. Many previous studies have used the approach of interpolating station data onto a grid and then conducting evaluations using the grid-to-grid method (Duan et al. 2016, Liu et al. 2015, Shukla et al. 2019). The present study adopted this approach to align with the commonly used practice.

The above explanation is now added in the revised manuscript in Line 263.

**13. Comment: L 261:** Evaluation can be included as a separate sub-heading as section 3.3

**Response:** 'Evaluation metrics' has been included as a separate sub-heading (2.4) in the revised manuscript.

**14. Comment**: **L 734:** The bullet points can be avoided in conclusion

**Response:** We thank the reviewer for the suggestion. Conclusion will be presented as a paragraph in the revised manuscript.

**15. Comment**: **L 747:** So the study can not generalise the use if any precipiattion data. Is ERA5 can be adopted to other similar agro-climatic conditions? What is the controbution of this study towards other data-sparse regions in India, other than the selected 4 locations?good to include future prospect.

**Response:** The following section on future prospects will be included in the revised manuscript:

The results of the study can be useful for the districts falling within the same agro-climatic regions. The state of the Tamil Nadu is divided into 7 agroclimatic zones. The present study includes 3 important agricultural zones. The results can be used for other districts within the same agroclimatic zones. The findings of this study are designed to support field-level experimentation and also provide a proof of concept for modelers developing climate data products, with the potential for extrapolation to other regions with similar agro-climatic conditions.

The above explanation is now added in the revised manuscript in Line 742 -L 749.

**Additional References:**

Arjune, S., & Kumar, V. S. (2023, February). Precision Agriculture: Influencing factors and challenges faced by farmers in delta districts of Tamil Nadu. In *2022 OPJU International Technology Conference on Emerging Technologies for Sustainable Development (OTCON)* (pp. 1-6). IEEE.

Balaganesh, G., Malhotra, R., Sendhil, R., Sirohi, S., Maiti, S., Ponnusamy, K., & Sharma, A. K. (2020). Development of composite vulnerability index and district level mapping of climate change induced drought in Tamil Nadu, India. *Ecological Indicators*, *113*, 106197.

Bosilovich, M. G., Lucchesi, R., and Suarez, M.: MERRA-2: File specification, 2015.

Duan, Z., Liu, J., Tuo, Y., Chiogna, G., and Disse, M.: Evaluation of eight high spatial resolution gridded precipitation products in Adige Basin (Italy) at multiple temporal and spatial scales, Science of the Total Environment, 573, 1536‑1553, https://doi.org/10.1016/j.scitotenv.2016.08.213, 2016.

Gardas, B. B., Raut, R. D., & Narkhede, B. (2018). Evaluating critical causal factors for post-harvest losses (PHL) in the fruit and vegetables supply chain in India using the DEMATEL approach. *Journal of cleaner production*, *199*, 47-61.

Government of Tamil Nadu: Statistical Handbook 2020-21: Sub chapter: Rainfall, 2022a.

IPCC, 2023: Summary for Policymakers. In: Climate Change 2023: Synthesis Report. Contribution of Working Groups I, II and III to the Sixth Assessment Report of the Intergovernmental Panel on Climate Change [Core Writing Team, H. Lee and J. Romero (eds.)]. IPCC, Geneva, Switzerland, pp. 1-34, doi: 10.59327/IPCC/AR6-9789291691647.001

Kanamitsu, M., Ebisuzaki, W., Woollen, J., Yang, S.-K., Hnilo, J. J., Fiorino, M., and Potter, G. L.: Ncep‑doe amip-ii reanalysis (r-2), Bull Am Meteorol Soc, 83, 1631‑1644, 2002.

LALMUANZUALA, B., SATHYAMOORTHY, N., KOKILAVANI, S., JAGADEESWARAN, R., & KANNAN, B. (2023). Drought analysis in southern region of Tamil Nadu using meteorological and remote sensing indices. *MAUSAM*, *74*(4), 973-988.

Liu, J., Duan, Z., Jiang, J., Zhu, A.X., 2015. Evaluation of three satellite precipitation products TRMM 3B42, CMORPH, and PERSIANN over a subtropical watershed in China. Adv. Meteorol. 2015 https://doi.org/10.1155/2015/151239.

Malaiarasan, U., Paramasivam, R., & Felix, K. T. (2021). Crop diversification: determinants and effects under paddy-dominated cropping system. *Paddy and Water Environment*, *19*, 417-432.

Paramasivam P (2023) Hundreds stranded as parts of India's Tamil Nadu flooded after heavy rain, *Reuters*: Hundreds stranded as parts of India's Tamil Nadu flooded after heavy rain | Reuters

Radhakrishnan, S., Duraisamy Rajasekaran, S. K., Sujatha, E. R., & Neelakantan, T. R. (2024). A Comparative Study on 2015 and 2023 Chennai Flooding: A Multifactorial Perspective. *Water*, *16*(17), 2477.

Shukla, A.K., Ojha, C.S.P., Singh, R.P., Pal, L., Fu, D., 2019. Evaluation of TRMM precipitation dataset over Himalayan Catchment: the upper Ganga Basin, India. Water (Switzerland) 11 (3). https://doi.org/10.3390/w11030613.

We thank the handling editor for providing valuable comments which, in our opinion has improved the quality of the manuscript. Here will outline all changes we have added to the previously revised manuscript. In particular, the following major change have been implemented:

- Vertical axis in Fig. 3 has been adjusted and the quality of the figure is now improved.
- Individual sub-figures have been labelled and rearranged in Fig. 5-8 to position the legends in one singular place.
- In Fig. 2, line type has been formatted for better visibility
- Title of Fig. 1 and Fig. 5-8 has now been revised.
- Included Table S1a in the supplementary file with the geographic coordinates of the ground stations.

We added a response to the Handling Editor's individual comments below.

1. **Comment:** Fig.1: The explanation of the horizontal and vertical alignment of groundstations is still not clear.

   **Response:** The stations were aligned into (an almost) regular grid by the state authorities/institutions for the study period (2003-14), which makes the districts selected "ideal" study areas for this evaluation study. The daily data was provided by the Public Works Department (PWD) of Tamil Nadu, which collects and maintains meteorological data for the entire state.
   The above explanation has been added to the Figure heading (L 177-179) in the revised manuscript. Further, the geographic coordinates of the ground stations has been included in the Supplementary document (Table S1a).

2. **Comment:** Fig 2 a & d. Line types are not consistent with the line type in the figure legend (e.g. Fig 2 a matches perfectly with PERSIANN-CDR, which is not possible).

   **Response:** As the differences between the station data and the precipitation product values for extreme precipitation were very small (less than 0.5 mm), the IDF curves overlapped.

   In the Coimbatore district, MSWEP and PERSIANN CDR had the closest precipitation intensity estimation with respect to the station data, as shown in Fig. 2a. The intensity of precipitation produced by station data was 16.056 mm/hour and 1.929 mm/hour for 1 hour and 24 hour duration, respectively. PERSIANN CDR produced 15.952 mm/hour and 1.917 mm/hour for 1 hour and 24 hour duration, respectively. In Madurai, the precipitation intensity of GPM-IMERG was very close to the station data. The intensity of precipitation produced by station data was 26.033 mm/hour and 3.128 mm/hour for 1 hour and 24 hour duration, respectively. GPM-IMERG produced 25.93 mm/hour and 3.117 mm/hour for 1 hour and 24 hour duration, respectively (Fig. 2b). In Tuticorin, ERA5-Land produced the closest precipitation intensity estimation with respect to the station data. Station data's precipitation intensity was 20.406 mm/hour and 2.452 mm/hour for 1 hour and 24 hour duration, respectively. ERA5-Land produced 20.269 mm/hour and 2.436 mm/hour for 1 hour and 24 hour duration, respectively (Fig. 2c). As the products produced closest estimate to the station data, the lines were closely plotted in the figure.

   To improve the visibility of products with similar patterns, the line types were adjusted in Fig. 2 a-d and the above explanation has been added in the Results section (L370 – L373, L374 -L 376, L381-L383).

3. **Comment:** Fig.3: The vertical axe titles are not uniformily edited, and often superseeds the grid values. The quality of these figures should be improved.

   **Response:** We thank the Editor for the suggestion. The figure has been improved in the revised manuscript (L 468 – L522).

4. **Comment**: Fig 5-8. Legend should positioned clearly at one singular place.

   **Response:** Based on the Editor's suggestion, legend position in Fig. 5- 8 has been altered to one singular place in the revised manuscript.

5. **Comment**: Fig. 5-8. The dots (circles) on the figures should be explained in detail.

   **Response:** In the station data spatial map, the distribution of ground station points and their respective linearly interpolated grids are plotted to understand the precipitation variation across the grids. The red circular dots represent the locations of ground stations from which precipitation data was collected for the period 2003–2014. The black stars indicate the linearly interpolated 0.1° grids. For the evaluation, only grids surrounded by at least one rain gauge were considered.
   The above explanation has been added to the Results (L 567 - L570) and Figure captions (Fig. 5-8) in the revised manuscript.